# Physics-Informed Pre-training on Efficient Electron-Density Images for Organic Material Property Prediction

**Zhixiang Cheng** [* 1 2] **Hongxin Xiang** [* 1 2] **Mingquan Liu** [3] **Tengfei Ma** [1 2] **Yingzhuo Tu** [1 2] **Wenjie Du** [4] **Bosheng Song** [1 2] **Yiping Liu** [1 5] **Xiangxiang Zeng** [1 2]

## Abstract

Precise property prediction of organic materials is pivotal for next-generation electronic and energy devices. In density functional theory (DFT), the electron density (ED) serves as the fundamental determinant of material properties. Yet, establishing it as an input modality for material property prediction has been impeded by two practical barriers: scarce large-scale ED data and the enormous computational complexity of ED representation. To bridge these gaps, we introduce VisionED, an efficient physics-informed model pre-trained on electron-density images. We curate a dataset of 2 million molecules and represent ED as multi-shot images that efficiently encode both geometric and electronic structure. VisionED is then pre-trained on 12 million multi-shot ED images via cross-scale, physics-informed pretext tasks. Empirical evaluations on photovoltaic and organic chromophore datasets show that VisionED outperforms state-of-the-art baselines by up to 27.0%, exhibiting superior robustness under distribution shifts and data scarcity. Notably, the model generalizes to unseen device-scale applications, successfully recovering experimental trends and mixing-ratio effects in ternary blends with an average accuracy of 92.77%. Moreover, relative to the previous ED point cloud, the ED image improves performance by 26.2% with 2.6× fewer memory and 4.6× lower time. The code and data are available at https://github.com/ZhixiangCheng/VisionED.

---
[*]Equal contribution [1]State Key Laboratory of Chemo and Biosensing, College of Computer Science and Electronic Engineering, Hunan University [2]Yuelushan Laboratory [3]Faculty of Health Sciences, University of Macau [4]School of Software Engineering, University of Science and Technology of China [5]Yuelushan Center for Industrial Innovation. Correspondence to: Xiangxiang Zeng <xzeng@hnu.edu.cn>.

*Proceedings of the 43$^{rd}$ International Conference on Machine Learning*, Seoul, South Korea. PMLR 306, 2026. Copyright 2026 by the author(s).

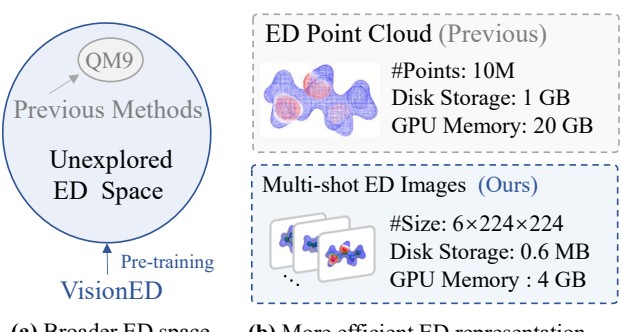

**(a)** Broader ED space  **(b)** More efficient ED representation

*Figure 1.* Prior ED methods were limited by **(a)** data scarcity and **(b)** high-dimensionality; VisionED enables large-scale pre-training via efficient ED images for robust prediction.

## 1. Introduction

Accelerating the discovery of organic materials is pivotal for advancements in energy storage, displays, and flexible electronics (Diesing et al., 2024; Yang et al., 2025). While traditional trial-and-error methods are prohibitively slow and costly (Li et al., 2022; Miao et al., 2025; Li et al., 2025b), data-driven AI has emerged as a transformative paradigm. By enabling high-throughput screening across the vast chemical space, machine learning (ML) models are shifting material design from empirical iteration to predictive precision (Butler et al., 2018; Zeni et al., 2025).

Despite this progress, existing paradigms predominantly rely on explicit representations, including hand-crafted descriptors (Sun et al., 2019a; Wu et al., 2024; Zhang et al., 2025; Liu et al., 2025), topology (Wang et al., 2022; Zeng et al., 2022), and geometric coordinates (Fang et al., 2022; Xiang et al., 2024), while overlooking the underlying electronic structure that fundamentally governs material properties. Consequently, such models frequently struggle with generalizability and accuracy (Sahu & Ma, 2019; Miyake & Saeki, 2021; Mahmood & Wang, 2021a; Oviedo et al., 2022), particularly when extrapolating to novel scaffolds or out-of-distribution (OOD) regimes.

To achieve physical fidelity, one must look to the foundational laws of quantum mechanics. According to Density Functional Theory (DFT) (Bartolotti & Flurchick, 1996;

Medvedev et al., 2017), the ground-state electron density (ED) uniquely determines all ground-state properties of a system. This raises a natural hypothesis: *can ED be established as an input modality for material property prediction?* In practice, however, making ED a scalable learning modality faces two key barriers. (i) *Data Scarcity:* Computing ED via DFT is computationally expensive. Historically, public ED resources were limited to small datasets like QM9 (130K samples) (Ramakrishnan et al., 2014; Jørgensen & Bhowmik, 2022), leaving the potential of large-scale ED pre-training largely unexplored (Figure 1a); (ii) *High-Dimensionality:* ED is a continuous volumetric field. As discretization resolution increases, the number of grid points can easily reach tens of millions, creating prohibitive bottlenecks in storage and GPU memory (Figure 1b). Together, these barriers have long impeded the effective integration of physics-informed ED into machine learning.

To overcome these challenges, we introduce VisionED, an efficient physics-informed model pre-trained on large-scale ED images. First, we construct a 12M ED images corpus, and concurrently compute ESSOR, a holistic suite of 69 descriptors across electronic, shape, electrostatic potential (ESP), orbital, and energy categories, to support physically grounded supervision. Inspired by the information compression strategies in DeepSeekOCR (Wei et al., 2025) and the observation from 3D reconstruction (Yao et al., 2018) that multi-shot images preserve rich spatial information, we represent ED as efficient multi-shot images to circumvent the computational costs of 3D volumetric data (Figure 1b). Finally, to incorporate physics-informed inductive biases, we pre-train VisionED on 12 million multi-shot ED images via cross-scale, physics-guided pretext tasks, learning representations that enforce multi-scale structural and physical consistency.

We summarize the main contributions as follows:

- We propose **VisionED**, a pre-training framework based on efficient multi-shot ED images for organic material property prediction.

- We curate a 12M DFT-computed ED images corpus and augment with a 69-dimensional ESSOR quantum descriptor.

- We design cross-scale, physics-guided pretext tasks to explicitly encode quantum and geometric inductive biases.

- We demonstrate that VisionED achieves state-of-the-art performance in organic material property prediction under various real-world challenging scenarios.

## 2. Related Work

**Organic Material Property Prediction.** Machine learning approaches for organic material property prediction broadly fall into two paradigms: descriptor-based and DL-based methods. The former relies on hand-crafted cheminformatic descriptors to map structures to properties (Sun et al., 2019b; Mahmood & Wang, 2021b; Wu et al., 2024; Zhang et al., 2025; Liu et al., 2025). Conversely, DL-based methods automate feature extraction directly from molecular representations, typically utilizing molecular graphs (Wang et al., 2023; Liao et al., 2024; Sun et al., 2024a) or sequences (Kuenneth & Ramprasad, 2023; Soares et al., 2024; Qiu et al., 2024). Departing from geometric topology or manual engineering approaches, we propose a novel physics-grounded ED representation to enable robust transfer across diverse material domains.

**Electron Density Representation Learning.** Research in this domain has predominantly focused on estimating electron density. Early approaches discretize space into voxels and regress densities with 3D U-Net (Sinitskiy & Pande, 2018). Subsequently, graph-based methods (Jørgensen & Bhowmik, 2020; 2022; Mitnikov & Jacobson, 2024) replace grids with message passing on atomic graphs and query points to predict $\rho(\mathbf{r})$, while recent neural operators treat estimation as learning mappings between function spaces (Kim & Ahn, 2024). Recently, Li et al. (2025a) views the electron density as a 3D grayscale image and uses a convolutional residual network to predict density. Conversely, utilizing electron density as an input modality remains nascent. Limited prior attempts include SepPC-NET (Wang et al., 2021), which consumes electrostatic potential point clouds for estrogenicity classification, and symmetry classification (Kim et al., 2024) models based on PointNet (Qi et al., 2017a). Departing from these task-specific applications, we establish ED image as an effective input modality and present the ED-derived pre-training model for material property prediction.

## 3. Preliminary

**Background.** Electron density is the probability of finding electrons at each point in space, and it shapes the electrostatics, orbital energies, and the optical and charge-transport behaviour of organic materials. Despite its central role, electron density remains underutilized as a first-class signal for material property prediction, largely due to scarce and costly annotations and the high-dimensional, inefficient representations. *These gaps motivate our ED dataset construction, efficient ED image representation, and the design of the VisionED framework.* We provide more background details in the Appendix C.1.

**Problem Definition.** We study the prediction of organic material properties from ED images. Given a molecule $m$ with 3D conformation $\mathcal{V}$, electron density $\rho_m(\mathbf{r})$, and electrostatic potential $V_m(\mathbf{r})$, we render a set of $K$ multi-shot ED images $\mathcal{I}_m = \{I_m^{(k)}\}_{k=1}^K$ as input, with $I_m^{(k)} \in$

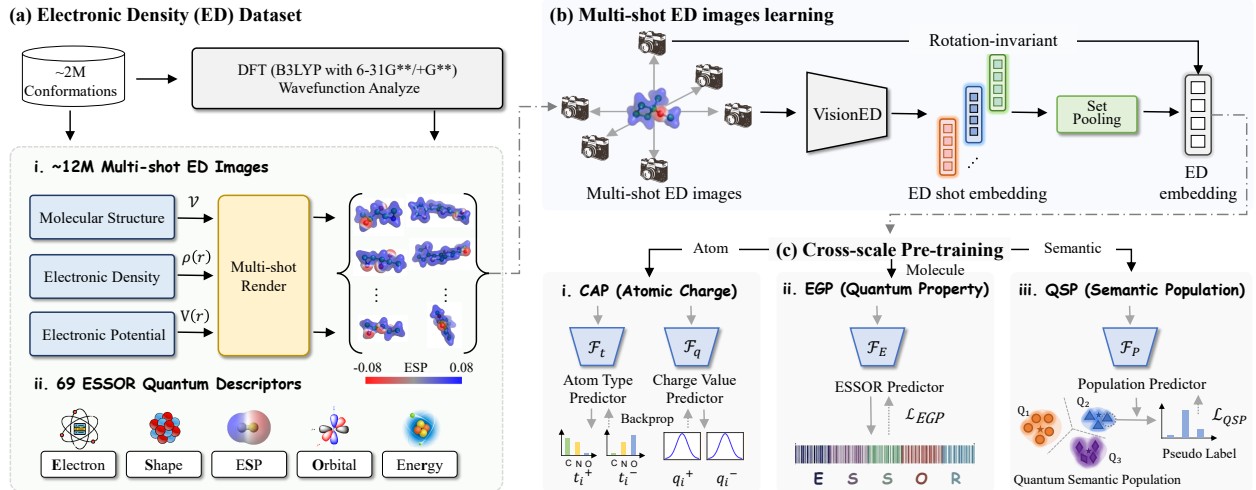

*Figure 2.* A diagram illustrating the VisionED framework. **(a)** We curate ∼2M molecular conformations with DFT-computed electronic density, and render multi-shot ED images from multiple viewpoints; in parallel, we compute 69 in-house ESSOR quantum descriptors. **(b)** A ViT-based encoder extracts per-view ED shot embeddings, which are aggregated by a set pooling module to produce a rotation-invariant molecular embedding. **(c)** We jointly optimize atom-level CAP, molecular-level EGP, and semantic-level QSP, enabling the model to align local electron, global quantum properties, and higher-level semantic signals.

$\mathbb{R}^{C \times H \times W}$, where $K = 6$ (top, bottom, left, right, front, and back), $C = 3$, $H = 224$, and $W = 224$ denote the number of shots, channels, height, and width, respectively. Our goal is to learn a model $f_\Theta$ that maps ED images to a target material property $\mathbf{y}_m \in \mathbb{R}^T$ denoted as $\hat{y}_m = f_\Theta(\mathcal{I}_m)$, where $T$ is the task dimension. For material pair, each sample is a pair $(m_a, m_d)$, and the model estimates the property $\hat{y}_{(m_d, m_a)} = f_\Theta(\mathcal{I}_{m_a}, \mathcal{I}_{m_d})$.

## 4. Method

### 4.1. Overview

Here, we propose the electron density image-derived framework (called **VisionED**). The overview of VisionED is illustrated in Figure 2, which is divided into 3 main modules: (a) *ED dataset construction* pipeline that collects ∼2M conformations, derives DFT-based ED/ESP, renders multi-shot ED images, and computes in-house ESSOR descriptors (Section 4.2); (b) *Multi-shot ED images learning* that encodes each shot with a shared ViT (Dosovitskiy et al., 2021) and aggregates shot features via set pooling to obtain an approximately rotation-invariant molecular embedding (Section 4.3); (c) *Cross-scale pre-training* scheme with atom, molecule, and semantic-level objectives to inject physical priors and improve transferability (Section 4.4).

### 4.2. ED Dataset Construction

**Electron density image generation.** To concisely describe ED image generation, Algorithm 1 summarizes a three-stage pipeline (Appendix C.2): (i) Leveraging the EDBench

database (Xiang et al., 2025), we select 2M unlabeled molecular geometries with their DFT-computed ED data, generated by Psi4 software (Turney et al., 2012; Smith et al., 2020); (ii) we load structures and cubes in PyMOL (DeLano et al., 2002), apply an ESP color ramp, and render ED isosurfaces at a fixed isovalue; and (iii) we perform multi-shot rendering by rotating the system along predefined axes to capture standardized ED images. Ultimately, this pipeline yields nearly 12M multi-shot ED images in total.

**ESSOR quantum descriptor calculation.** To provide a comprehensive and physically meaningful representation of molecular quantum properties, we propose a descriptor set named **ESSOR**, which encompasses five categories, including **E**lectronic (n=19), **S**hape (n=11), E**S**P (n=20), **O**rbital (n=14), and Ene**r**gy descriptors (n=5), resulting in a total of 69 descriptors that reflect the molecular electronic environment, geometric structure, and quantum attributes. Among them, energy and orbital descriptors are directly extracted from the outputs of Psi4 during DFT calculation, while the rest are computed using Multiwfn (Lu & Chen, 2012), an open-source wavefunction analysis tool (Appendix C.3). We further apply a unified standardization scheme to all descriptors to ensure consistent scaling across molecules.

### 4.3. Multi-shot ED Images Learning

We propose a *Multi-shot ED Images Learning* strategy that projects high-dimensional ED into efficient image representations compatible with standard vision backbones. Crucially, by aggregating views from diverse angles, this paradigm not only compensates for the information loss

inherent in single-shot projections to recover holistic 3D spatial structures but also endows the model with approximate rotation invariance via group averaging.

Formally, given a molecule $m$ with a set of $K$ multi-shot ED images $\mathcal{I}_m = \{I_m^{(k)}\}_{k=1}^K$. We feed images into a VisionED encoder $f_E$ with patch size of $n_p = 16$ to obtain shot-level embedding:

$$\mathbf{h_k} = f_E(\mathcal{I}_m^{(k)}), \quad \mathbf{h_k} \in \mathbb{R}^{n_t^2 \times d}, \qquad (1)$$

where $d$ is the dimension, $n_t = H/n_p = W/n_p$. For all shot embeddings, we perform average pooling to get the ED representation:

$$\mathbf{h}_m = \text{Pool}_{\text{set}}\left(\{\mathbf{h}_k\}_{k=1}^K\right), \mathbf{h}_m \in \mathbb{R}^d. \qquad (2)$$

**Theoretical Analysis.** We substantiate the claimed approximate rotation invariance from a group-theoretic perspective. Let $G = \text{SO}(3)$ act on the input space via $(g, x) \mapsto T_g x$. The multi-shot pooling can be viewed as an empirical estimator of the group-averaged representation defined by $\widehat{\Phi}_K(x) := \frac{1}{K} \sum_{k=1}^K \phi(T_{g_k} x)$. As the shots $\{g_k\}_{k=1}^K$ cover the manifold $G$, $\widehat{\Phi}_K$ converges to a Haar integral, thereby satisfying approximate global rotation invariance: $\widehat{\Phi}_K(T_h x) \approx \widehat{\Phi}_K(x), \forall h \in G$. See Appendix A for formal proofs and error bounds.

### 4.4. Cross-scale Pre-training of VisionED

To effectively encode electron density representations with physics-informed inductive biases, we introduce three cross-scale pre-training objectives targeting atomic charges, molecular quantum properties, and semantic signal.

**Charge-Aware pre-training (CAP).** To encourage VisionED to capture localized charge patterns, we introduce CAP, which supervises the model to predict the most positively and negatively charged atoms (reactive electrophilic and nucleophilic centers) from ED images. For a material molecule $m$ with Mulliken charges $\{q_i\}$ computed by DFT, we select the charge extrema $q^+ = \max(q_i)$ and $q^- = \min(q_i)$ and their corresponding atom types $t^+, t^- \in \{1, \ldots, U\}$, with $U$ being the total number of element classes. CAP combines (i) atom-type classification for $(t^+, t^-)$ and (ii) charge-value regression for $(q^+, q^-)$. The objective is

$$\mathcal{L}_{\text{CAP}} = \arg\min_\theta \frac{1}{N} \sum_{i=1}^N [\ell_1(\hat{t}_i, t_i) + \ell_2(\hat{q}_i, q_i)], \quad (3)$$

where $(t_i, q_i) \in \{(t_i^+, q_i^+), (t_i^-, q_i^-)\}$, and $(\hat{t}_i, \hat{q}_i) = f_\theta(\mathcal{I}_m)$ are predictions from the encoder $f_\theta$ with parameters $\theta$, $\ell_1$ is the focal loss function, $\ell_2$ is the Smooth L1 loss function.

**ESSOR-Guided pre-training (EGP).** To explicitly link electron density to quantum molecular properties, we design EGP, which trains the model to regress our in-house ESSOR descriptors from multi-view ED images. Given ED images $I_m^{(k)} \in \mathbb{R}^{C \times H \times W}$ for molecule $m$, the encoder $f_\theta$ predicts an ESSOR vector $Q_i \in \mathbb{R}^{d_E}$ (with $d_E=69$ in our setting). We optimize an element-wise regression objective:

$$\mathcal{L}_{\text{EGP}} = \arg\min_\theta \frac{1}{N} \sum_{i=1}^N \ell(f_\theta(I_m), Q_i), \qquad (4)$$

where $\ell$ denotes the regression loss.

**Quantum semantic population (QSP).** To encourage global semantic awareness beyond scalar regression, we introduce QSP, which learns *quantum semantic populations* by grouping molecules with similar electronic behaviors. We first compute the full ESSOR descriptors $\mathbf{s}_i \in \mathbb{R}^d$ for each molecule, and obtain $P$ prototype centroids $\{\boldsymbol{\mu}_p\}_{p=1}^P$ via $k$-means (we use $P=100$). Each molecule is assigned a pseudo-label

$$y_i = \arg \min_{p \in \{1, \ldots, P\}} \|\mathbf{Q}_i - \boldsymbol{\mu}_p\|_2, \qquad (5)$$

Given multi-shot ED images $\mathcal{I}_m$, the encoder $f_\theta$ predicts a categorical distribution over prototypes. We train QSP with a cross-entropy objective:

$$\mathcal{L}_{\text{QSP}} = \arg\min_\theta \frac{1}{N} \sum_{i=1}^N \ell(f_\theta(I_m), y_i). \qquad (6)$$

The total cost function $\mathcal{L}$ is defined as

$$\mathcal{L} = \lambda_1 \mathcal{L}_{CAP} + \lambda_2 \mathcal{L}_{EGP} + \lambda_3 \mathcal{L}_{QSP}, \qquad (7)$$

where $\lambda_1$, $\lambda_2$, and $\lambda_3$ are the balance coefficient. We describe more details of pre-training and ablation study of the pretext task in Appendix D.

### 4.5. Fine-tuning Process

To fine-tune VisionED for downstream molecular tasks, we adopt a pairwise input format (e.g., acceptor-donor or emitter-solvent). For each shot $k \in \{1, \ldots, K\}$, an ED image pair $(a_i^k, d_i^k)$ is independently encoded by a shared ViT encoder $f_\theta$, yielding latent features $\mathbf{h}(a_i^k), \mathbf{h}(d_i^k) \in \mathbb{R}^d$. We concatenate the pairwise features and feed them to an MLP head $\omega$ to obtain shot-level logits $\mathbf{l}_i^k = \omega([\mathbf{h}(a_i^k); \mathbf{h}(d_i^k)]) \in \mathbb{R}^T$, where $T$ is the task dimension. We then average logits over all shots to form the final prediction, $\mathbf{l}_i = \frac{1}{K} \sum_{k=1}^K \mathbf{l}_i^k$. For regression, we optimize the Smooth L1 loss.

## 5. Experiments

In this section, to evaluate the effectiveness of VisionED, we carefully consider the following key *research questions*:

*Table 1.* The mean absolute error (MAE) performance comparison on the OPEP2 dataset under acceptor and donor scaffold splits. Results are reported as the mean of three independent runs, with standard deviations in parentheses. The **bold** and underlined values indicate the best and second-best performance, respectively. $\Delta$ denotes the relative gain of VisionED over the leading baseline.

| Model | Acceptor scaffold split | | | | Donor scaffold split | | | |
| --- | --- | --- | --- | --- | --- | --- | --- | --- |
| | $V_{OC} \downarrow$ [V] | $J_{SC} \downarrow$ [mA·cm$^{-2}$] | FF $\downarrow$ [%] | PCE $\downarrow$ [%] | $V_{OC} \downarrow$ [V] | $J_{SC} \downarrow$ [mA·cm$^{-2}$] | FF $\downarrow$ [%] | PCE $\downarrow$ [%] |
| SVM | 0.086 | 4.451 | 8.255 | 2.994 | 0.087 | 4.063 | 10.214 | 3.546 |
| RF | 0.079(0.002) | 3.781(0.087) | 8.802(0.472) | 2.460(0.068) | 0.097(0.007) | 3.917(0.147) | 10.081(0.020) | 3.189(0.127) |
| X-3D | 0.100(0.002) | 5.302(0.064) | 8.433(0.159) | 3.412(0.086) | 0.091(0.003) | 4.997(0.039) | 9.685(0.108) | 3.510(0.154) |
| PointVector | 0.103(0.002) | 5.687(0.077) | 8.528(1.149) | 3.655(0.154) | 0.101(0.005) | 5.007(0.161) | 9.701(0.094) | 3.730(0.358) |
| GeoFormer | 0.079(0.002) | 3.740(0.046) | 8.002(0.026) | 2.458(0.054) | 0.076(0.001) | 3.887(0.142) | 9.243(0.256) | 3.240(0.015) |
| EquiformerV2 | 0.077(0.002) | 3.653(0.023) | 7.913(0.014) | 2.530(0.063) | 0.075(0.001) | 3.675(0.174) | 9.052(0.123) | 3.175(0.013) |
| ImageMol | 0.080(0.004) | 3.879(0.019) | 8.324(0.440) | 2.658(0.192) | 0.083(0.006) | 3.912(0.182) | 9.078(0.136) | 3.203(0.114) |
| VideoMol | 0.078(0.002) | 3.869(0.073) | 7.787(0.103) | 2.448(0.044) | 0.076(0.003) | 3.640(0.055) | 9.047(0.149) | 3.125(0.107) |
| VisionED | **0.075**(0.002) | **3.518**(0.034) | **7.541**(0.107) | **2.271**(0.046) | **0.073**(0.001) | **3.569**(0.122) | **8.891**(0.116) | **3.034**(0.018) |
| $\Delta$ | ↑ 2.6% | ↑ 6.4% | ↑ 3.2% | ↑ 7.2% | ↑ 2.7% | ↑ 2.0% | ↑ 1.7% | ↑ 2.9% |

*Table 2.* The MAE performance comparison of photovoltaic property prediction under 5-fold cross-validation.

| Model | 5-fold cross-validation | | | |
| --- | --- | --- | --- | --- |
| | $V_{OC} \downarrow$ | $J_{SC} \downarrow$ | FF $\downarrow$ | PCE $\downarrow$ |
| SVM | 0.116(0.016) | 2.849(0.153) | 9.303(0.523) | 1.486(0.097) |
| RF | 0.097(0.015) | 2.810(0.248) | 8.443(0.982) | 1.495(0.093) |
| X-3D | 0.138(0.019) | 3.868(0.298) | 10.087(0.927) | 1.800(0.136) |
| PointVector | 0.121(0.013) | 4.095(0.419) | 10.599(0.678) | 1.986(0.093) |
| GeoFormer | 0.111(0.015) | 3.917(0.248) | 8.005(0.642) | 1.404(0.059) |
| EquiformerV2 | 0.098(0.014) | 2.820(0.045) | 7.945(0.782) | 1.442(0.052) |
| ImageMol | 0.106(0.010) | 3.016(0.227) | 7.803(1.002) | 1.504(0.082) |
| VideoMol | 0.102(0.009) | 2.750(0.280) | 8.255(0.930) | 1.437(0.148) |
| VisionED | **0.094**(0.012) | **2.339**(0.287) | **7.404**(1.096) | **1.278**(0.098) |
| $\Delta$ | ↑ 3.1% | ↑ 14.9% | ↑ 5.1% | ↑ 9.0% |

**Q1 (Robustness)**: Does VisionED exhibit superior performance on organic material property prediction across various real-world challenging scenarios? **Q2 (Effectiveness)**: Is the proposed multi-shot ED image representation a sufficient and effective alternative to ED point cloud? **Q3 (Interpretability)**: Does VisionED provide meaningful and physically grounded explanations for its predictions?

### 5.1. Experimental Settings

**Datasets and evaluation protocol.** We pretrain VisionED on 2M unlabeled molecules with DFT-computed ED from EDBench and 69 in-house computed ESSOR quantum descriptors, rendering multi-shot ED images per molecule. The pre-training dataset is randomly split into 80%/10%/10% for train/valid/test. We evaluate on three downstream benchmarks (Appendix B.1) for organic photovoltaic and optoelectronic materials: HOPV15 (Lopez et al., 2016), OPEP2 (Greenstein & Hutchison, 2023), and

Deep4Chem (Joung et al., 2020). HOPV15 contains experimental PCEs for donor-acceptor (D-A) pairs; OPEP2 is a larger OPV benchmark with 1,000 D-A combinations and experimental PCEs; Deep4Chem is a large experimentally curated photophysical dataset with absorption, emission, PLQY, and FWHM measurements spanning diverse emitter-solvent systems. To better mimic practical design, we use scaffold-based split; for HOPV15, we adopt 5-fold cross-validation. We report mean absolute error (MAE), averaged over three random seeds.

**Baselines.** To ensure a comprehensive evaluation, we benchmark VisionED against a diverse array of methods ranging from traditional machine learning to state-of-the-art deep learning architectures (Appendix B.3). Specifically, we selected representative baselines across four distinct categories: (i) Traditional ML Baselines: **SVM** (Zhang, 2001) and **RF** (Breiman, 2001) using ECFP fingerprints (Rogers & Hahn, 2010); (ii) Point Cloud Models adapted for ED point cloud representation: **PointVector** (Deng et al., 2023) and **X-3D** (Sun et al., 2024b); (iii) Geometric Graph Models: **GeoFormer** (Wang et al., 2023) and **EquiformerV2** (Liao et al., 2024); (iv) Visual Molecular Models: **ImageMol** (Zeng et al., 2022), which treats molecules as static 2D images, and **VideoMol** (Xiang et al., 2024), which captures 3D dynamics via geometric video.

**Implementation details.** The Encoder of VisionED is built on ViT-Base/16. In the pre-training stage, we set a learning rate of 0.005, a batch size of 256, a momentum of 0.9, and a weight decay of 1e-4 for pre-training. We pre-trained VisionED over 50 epochs on a server with Intel(R) Xeon(R) Platinum 8375C@2.90GHz CPU and 8 NVIDIA 4090 (24GB) GPUs, which took about 4 days. For downstream adaptation, we append an MLP head to the encoder and perform a grid search over task-specific hyperparame-

*Table 3.* The MAE performance comparison of optical property prediction under emitter and solvent scaffold split for the Deep4Chem.

| Model | Emitter scaffold split | | | | Solvent scaffold split | | | |
|---|---|---|---|---|---|---|---|---|
| | $\lambda_{\text{Abs}} \downarrow$ [nm] | $\lambda_{\text{Emi}} \downarrow$ [nm] | PLQY $\downarrow$ [-] | FWHM $\downarrow$ [nm] | $\lambda_{\text{Abs}} \downarrow$ [nm] | $\lambda_{\text{Emi}} \downarrow$ [nm] | PLQY $\downarrow$ [-] | FWHM $\downarrow$ [nm] |
| SVM | 73.078 | 80.230 | 0.297 | 20.787 | 92.729 | 70.734 | 0.295 | 22.438 |
| RF | 61.333(0.662) | 69.230(0.483) | 0.303(0.007) | 24.434(0.940) | 30.827(0.497) | 53.770(1.926) | 0.258(0.013) | 25.705(1.415) |
| X-3D | 81.780(1.155) | 87.074(0.698) | 0.299(0.002) | 27.841(0.185) | 70.773(2.111) | 87.918(9.490) | 0.283(0.004) | 27.868(0.120) |
| PointVector | 83.019(1.749) | 81.699(1.826) | 0.298(0.004) | 29.816(0.273) | 86.183(2.111) | 86.183(9.520) | 0.293(0.003) | 29.185(0.347) |
| GeoFormer | 61.719(1.739) | 65.580(0.294) | 0.267(0.002) | 21.331(0.243) | 39.134(0.652) | 47.546(4.123) | 0.224(0.002) | 20.163(0.741) |
| EquiformerV2 | 58.005(1.458) | 65.372(0.241) | 0.260(0.003) | 20.756(0.452) | 31.476(0.754) | 45.742(3.813) | 0.218(0.001) | 18.421(0.145) |
| ImageMol | 67.910(0.688) | 72.992(2.603) | 0.268(0.003) | 22.215(0.453) | 52.284(2.111) | 70.466(9.490) | 0.232(0.001) | 21.430(0.335) |
| VideoMol | 59.573(0.149) | 64.395(0.318) | 0.255(0.002) | 20.660(0.186) | 32.045(1.477) | 52.926(1.152) | 0.222(0.006) | 19.153(0.142) |
| VisionED | **51.023**(0.739) | **60.653**(0.192) | **0.244**(0.004) | **19.596**(0.300) | **22.502**(0.580) | **39.134**(3.206) | **0.208**(0.004) | **17.319**(0.211) |
| $\Delta$ | ↑12.0% | ↑5.8% | ↑4.3% | ↑5.2% | ↑27.0% | ↑14.4% | ↑4.6% | ↑6.0% |

ters on the validation set (Appendix B.2).

## 5.2. Performance of VisionED (Q1)

In real-world materials discovery, molecules encountered at inference time frequently deviate from the training distribution due to unseen scaffolds and functional substitutions, inducing pronounced distribution shifts. Conventional models often degrade under such OOD conditions. Meanwhile, acquiring large-scale labels is costly and slow, making data scarcity pervasive. Motivated by these practical challenges, we respond to **Q1** by evaluating VisionED under (1) *scaffold split*, (2) *OOD test*, (3) *low-data*, (4) *novel materials*.

### 5.2.1. SCAFFOLD SETTING

To better reflect real-world molecular design scenarios, we adopted scaffold-based splits where training and test molecules share minimal structural overlap. Tables 1 reports MAE on OPV property prediction for the OPEP2 dataset under scaffold evaluation. VisionED achieves the lowest error under both *acceptor* and *donor* scaffold splits. For the *acceptor* split, VisionED attains MAE of 3.518 ($J_{\text{SC}}$), and 2.271 (PCE), yielding up to 7.2% relative improvement on PCE and 6.4% on $J_{\text{SC}}$ over the strongest baseline. Under the *donor* split, VisionED remains robust and delivers the best MAE across all key properties, with consistent gains (up to 2.7% on $V_{\text{OC}}$). To further evaluate model performance, we tested VisionED on the HOPV15 dataset in Table 2. Using 5-fold cross-validation on the small HOPV15 benchmark, VisionED again achieves the best MAE on all four properties, with an average 8.0% improvement ranging from 5.1% to 14.9% compared with the SOTA baseline.

We next evaluated VisionED on optical property prediction tasks using the Deep4Chem dataset. Under *emitter* scaffold split, VisionED achieved a notable MAE reduction relative to prior models: for example, an MAE of 51.023±0.739

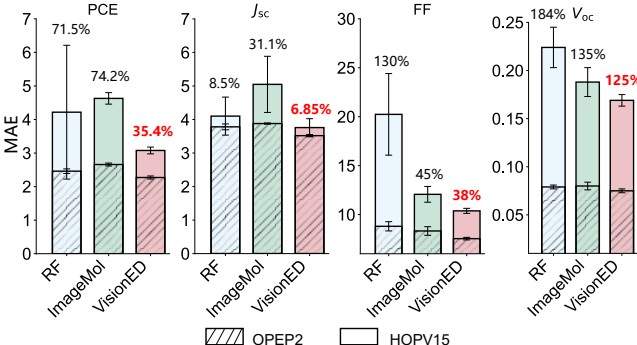

*Figure 3.* OOD evaluation of VisionED. Dashed bars indicate in-distribution results on the OPEP2 test set. Red text above the bars indicates the percentage decrease in MAE compared to the in-distribution OPEP2 test set.

nm for $\lambda_{\text{Abs}}$, which outperformed VideoMol and RF, along with similar gains in $\lambda_{\text{Emi}}$, PLQY, and FWHM. In *solvent* scaffold split, VisionED achieved the lowest MAE, with an average 13.0% improvement ranging from 4.6% to 27.0% compared with VideoMol.

### 5.2.2. OOD SETTING

We assessed VisionED's generalization in an OOD setting by training models on the OPEP2 dataset and testing on HOPV15, which contains structurally distinct compounds, as illustrated by t-SNE embedding (See Appendix Figure 7). In this cross-dataset transfer, VisionED significantly outperformed baseline methods across all four metrics with an average 44.7% improvement compared with ImageMol and an average 47.4% improvement compared with RF (Figure 3). Notably, compared to its in-distribution performance, VisionED exhibited the smallest drop in accuracy when transferred to HOPV15, only a 6.8% increase in $J_{\text{SC}}$ MAE, demonstrating remarkable robustness to domain shift.

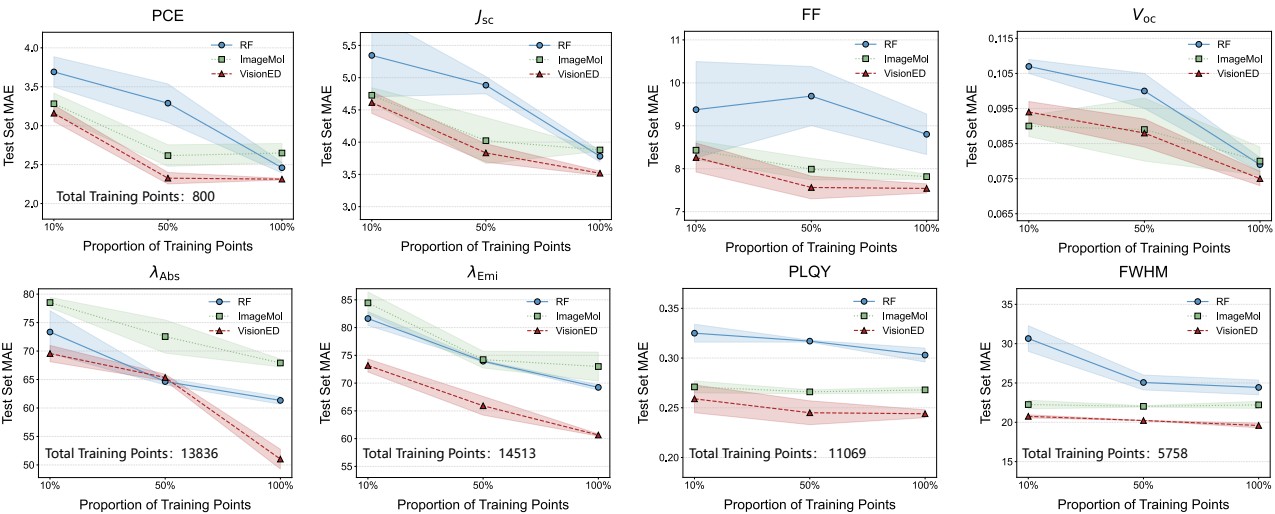

*Figure 4.* Low data evaluation of VisionED. Performance on photovoltaic (**top**) and optical (**down**) property prediction with varying proportions (10%, 50%, 100%) of training data, indicating sensitivity to data availability.

### 5.2.3. LOW DATA SETTING

Next, we evaluated performance under limited training data. Using 10%, 50%, and 100% of available training points, we tested models on test sets for both photovoltaic and optical tasks (Figure 4). Across all settings, VisionED consistently outperformed RF and ImageMol. For instance, with only 50% of the training data, VisionED reduced MAE in PCE by 11.1% relative to ImageMol and 29.3% relative to RF. Remarkably, under the 50% training data condition, VisionED (MAE=2.326±0.074) even surpasses the performance of baseline models trained with 100% of the data (MAE=2.649±0.123), highlighting its exceptional sample efficiency and generalization capacity. Similar trends held for optical properties, where VisionED maintained strong performance in predicting $\lambda_{\text{Abs}}, \lambda_{\text{Emi}}$, PLQY, and FWHM even with minimal supervision.

### 5.2.4. NOVEL MATERIALS SYSTEMS

We first employed VisionED to conduct large-scale virtual screening across a vast, unexplored donor-acceptor space (See Appendix D.6 and Figure 10). VisionED identified the unseen D18:AQx-2 pair from 125,790 candidates, yielding a predicted PCE of 17.15% that closely matches the value recently reported in experimental literature (17.20%) (Jiang et al., 2023).

Extending beyond binary systems, we challenge VisionED with a strictly out-of-distribution task: compositional extrapolation to ternary systems. While VisionED is trained exclusively on binary D-A pairs, ternary blends (involving two donors or two acceptors) represent a higher-order complexity critical for high-performance OPVs (Guan et al., 2024; Sun et al., 2024c). To verify performance on these

*Table 4.* Performance of VisionED on unseen ternary D-A combinations. For the PCE, prediction accuracy is calculated as $1 - \delta$, where $\delta$ is the deviation between predicted and experimental values following Zhang et al. (2025). The experimental values are from their original papers.

| Index | D-A Combination | $\text{PCE}_{\text{pred}}$ | Accuracy |
|---|---|---|---|
| **$D_1$+$D_2$:A** ternary combinations | | | |
| 1 | D18+PTQ10:m-BTP-C6Ph | 17.62(0.96) | 95.92(3.27) |
| 2 | PBDB-TF+PBDB-T-SF:Y6 | 16.86(0.52) | 97.32(3.17) |
| 3 | D18+PBDB-TF:Y6 | 16.60(0.71) | 88.22(4.78) |
| 4 | J71+PTB7-Th:IHIC | 9.14(1.34) | 88.43(3.55) |
| 5 | PBDB-T+PTB7-Th:IHIC | 8.55(1.53) | 86.48(6.19) |
| **D:$A_1$+$A_2$** ternary combinations | | | |
| 6 | PBDB-TF:Y6+SY3 | 17.12(0.44) | 98.08(5.19) |
| 7 | PBDB-TF:Y6+BTTPC | 17.62(0.92) | 95.44(5.44) |
| 8 | PBDB-TF:Y6+IT-4F | 15.77(0.27) | 96.77(4.81) |
| 9 | PBQx-TF:eC9-2Cl+F-BTA3 | 18.66(1.22) | 95.44(3.84) |
| 10 | PBDB-TF:L8-BO+BTP-eC9 | 17.59(0.61) | 90.89(3.16) |
| 11 | PTZ1:IDIC+ITIC | 9.97(0.98) | 93.41(6.43) |
| 12 | PBDB-TF:Y6+PC71BM | 16.07(0.73) | 95.54(3.26) |
| 13 | PTB7-Th:Y6+PC71BM | 11.45(0.45) | 80.13(4.69) |
| 14 | PTQ10:Y6+PC71BM | 15.57(1.04) | 94.00(1.53) |
| 15 | PBDB-T-2Cl:Y6+PC71BM | 15.95(0.55) | 95.47(3.28) |

*novel material states*, we model ternary blends as ratio-weighted ensembles of their constituent binary interactions without fine-tuning. We validate this zero-shot capability on five $D_1$+$D_2$:A (Zhu et al., 2022; Chang et al., 2020) and ten D:$A_1$+$A_2$ (Liu et al., 2020; Wang et al., 2020; An et al., 2019) configurations (Table 4). Remarkably, VisionED

*Table 5.* Efficiency comparison between ED point clouds and ED images on HOPV15 PCE prediction under the same experimental setting. Point cloud uses PointVector (Deng et al., 2023) and Image uses a ViT-Base encoder. Pre-processing time refers to the farthest point sample (Qi et al., 2017b) of point clouds and rendering of ED images. Total time is the sum of pre-processing and model training time. Generation of the 60-shot ED images: 20 uniform samples were taken along each of the three orthogonal axes (x, y, and z).

| Modality | #Num | PCE | GPU memory (MiB) | Disk storage (MB) | Pre-process time (s) | Total time (s) |
|---|---|---|---|---|---|---|
| Point cloud | 512 | $2.031_{(0.121)}$ | 5,661 | 2.7 | ∼665 | ∼915 |
| | 1024 | $2.001_{(0.103)}$ | 11,011 | 5.4 | ∼1,254 | ∼1,545 |
| | 2048 | $\mathbf{1.986_{(0.093)}}$ | 13,313 | 10.8 | ∼5,357 | ∼5,694 |
| | 4096 | $2.038_{(0.122)}$ | 20,757 | 21.5 | ∼6,594 | ∼7,050 |
| Image | 1 | $1.596_{(0.159)}$ | 4,746 | 3.5 | ∼218 | ∼547 |
| | 6 | $\mathbf{1.465_{(0.081)}}$ | 4,746 | 35 | ∼425 | ∼1,243 |
| | 60 | $1.450_{(0.046)}$ | 4,746 | 360 | ∼3,620 | ∼7,920 |

achieves an average PCE accuracy of 92.77%. Crucially, as shown in Figure 5, VisionED demonstrates high sensitivity to compositional variations: it accurately recovers experimentally observed optimal mixing ratios (e.g., peak performance at 15% for PBDB-TF+PBDB-T-SF:Y6) and captures the asymmetric efficiency decay in non-optimal regimes.

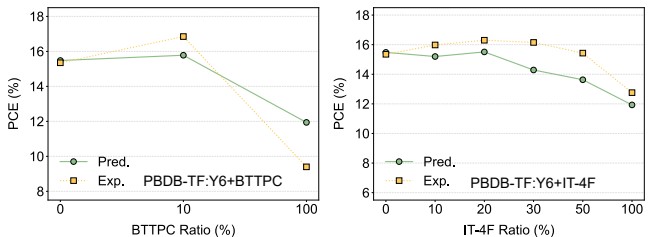

*Figure 5.* Predicted and experimental PCE values across different component ratios in ternary blends.

### 5.3. Efficiency Analysis (Q2)

Table 5 shows that ED images achieve a better accuracy and efficiency than ED point clouds. For point clouds, increasing #Num substantially raises memory (5.7→20.8 GiB) and pre-processing time (665→6,594 s), while MAE does not improve monotonically. Under a comparable GPU memory budget (∼5 GiB), ED images exhibit higher information density. The 512-point PointVector baseline consumes 5.7 GiB yet attains a PCE MAE of 2.031, whereas our 6-shot image model uses lower memory and achieves a much lower MAE of 1.465, i.e., a 27.9% error reduction. Compared to the best point cloud setting, 6-shot further improves MAE by 26.2% with 2.6× less memory and 4.6× lower total time.

Furthermore, we analyze the intra-modality scaling to justify our shot selection. Increasing from 1-shot to 6-shot substantially improves accuracy, corresponding to an 8.2% relative improvement. In contrast, scaling from 6-shot to 60-shot yields only marginal gains (only 1% improvement). Meanwhile, the computational and storage overhead grows sharply (∼10×). Therefore, we adopt 6-shot as a practical

trade-off that captures most of the performance benefit with manageable cost.

### 5.4. Attention Interpretability (Q3)

Attention visualizations corroborate these findings that VisionED leverages physically grounded cues from electron density to drive accurate and interpretable predictions (Figure 6). For an emitter, VisionED primarily attends to the planar aromatic core and conjugated bridges that dominate $\pi \rightarrow \pi^*$ transitions (Yamaguchi et al., 2008; Gong et al., 2007). For an acceptor, attention concentrates around electron-withdrawing heteroatoms (e.g., N/O), while for a donor it emphasizes regions with more positive ESP, consistent with donor-acceptor electrostatics (D'Avino et al., 2016). In contrast, ImageMol and VideoMol largely focus on coarse molecular contours, showing weaker sensitivity to electron variations.

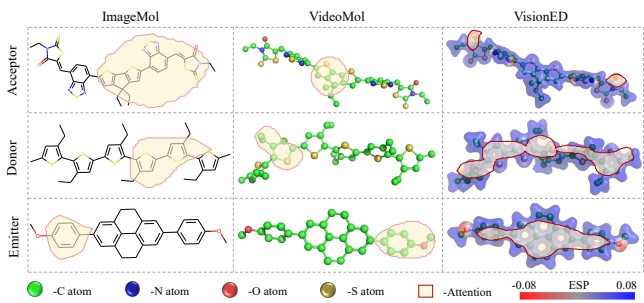

*Figure 6.* Attention visualizations compared with baselines.

## 6. Conclusion

In this work, we introduced VisionED, the first foundation model that establishes electron density as a scalable and primary input modality for organic material analysis. Extensive empirical evaluations demonstrate its superiority across various real-world scenarios. Despite VisionED's potential in AI-based material discovery, several limitations remain to be addressed. These limitations are primarily in

these aspects: (1) the generation of ED image is constrained by the reliance on DFT; (2) the representation of dynamic processes is absent; (3) the number of downstream tasks is limited; (4) the capability for de novo design is limited. Future works are detailed in the Appendix E.

## Impact Statement

This work advances the field of AI for materials science by demonstrating the feasibility and efficacy of learning directly from fundamental quantum variables. VisionED has the potential to significantly accelerate the discovery of high-performance organic materials for next-generation energy and display technologies. However, it should also be approached with caution due to the potential misuse in the development of hazardous products.

## Acknowledgments

This work was supported by Yuelushan Laboratory Breeding Program (Grant No. YLS-2026-ZY01001), Fundamental and Interdisciplinary Disciplines Breakthrough Plan of the Ministry of Education of China (Grant No. JYB2025XDXM602), the National Natural Science Foundation of China (Grant No. 62425204; U22A2037; 62450002; 62432011; 625B2067; 62522110 and 62472152), the Hunan Provincial Natural Science Foundation of China (Grant No. 2024JJ4015), and the Project of Yuelushan Center for Industrial Innovation (Grant No. 2025YCII0214).

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

# A. Theoretical Analysis

Let $G = \mathrm{SO}(3)$ be the 3D rotation group equipped with the normalized Haar measure $\mu$ (which is bi-invariant on compact groups). Assume $G$ acts measurably on an input space $\mathcal{X}$ via $(g, x) \mapsto T_g x$. Let $\phi : \mathcal{X} \to \mathbb{R}^d$ be a measurable encoder.

Define the Haar-averaged representation

$$\Phi(x) := \int_G \phi(T_g x) \, d\mu(g) \in \mathbb{R}^d. \tag{A.1}$$

Given $K$ shots $g_1, \ldots, g_K \in G$, define the empirical multi-shot mean

$$\widehat{\Phi}_K(x) := \frac{1}{K} \sum_{k=1}^{K} \phi(T_{g_k} x) \in \mathbb{R}^d. \tag{A.2}$$

**Theorem A.1** (Exact invariance of Haar averaging). *For any $h \in G$ and any $x \in \mathcal{X}$, $\Phi(T_h x) = \Phi(x)$.*

*Proof.* By definition,

$$\Phi(T_h x) = \int_G \phi(T_g T_h x) \, d\mu(g) = \int_G \phi(T_{gh} x) \, d\mu(g). \tag{A.3}$$

Let $g' = gh$. Since $\mu$ is (right-)invariant under group translation (bi-invariant on SO(3)), $d\mu(g) = d\mu(g')$, hence

$$\Phi(T_h x) = \int_G \phi(T_{g'} x) \, d\mu(g') = \Phi(x). \tag{A.4}$$

$\square$

**Randomized view sampling.** The following concentration result assumes that the shots are sampled i.i.d. from the Haar measure. This corresponds to Monte-Carlo integration of the Haar average.

**Lemma A.2** (Concentration of the empirical group average). *Fix $x \in \mathcal{X}$ and assume $\|\phi(T_g x)\|_2 \leq B$ for all $g \in G$. If $g_1, \ldots, g_K \overset{i.i.d.}{\sim} \mu$, then for any $\varepsilon > 0$,*

$$\mathbb{P}\left( \left\| \widehat{\Phi}_K(x) - \Phi(x) \right\|_2 \geq \varepsilon \right) \leq 2d \exp\left( -\frac{K\varepsilon^2}{2B^2 d} \right). \tag{A.5}$$

*Proof.* For each coordinate $j \in \{1, \ldots, d\}$, define $Z_{k,j} := \phi_j(T_{g_k} x)$. Since $\|\phi(T_{g_k} x)\|_2 \leq B$, we have $|Z_{k,j}| \leq B$. Moreover, $\mathbb{E}[Z_{1,j}] = \int_G \phi_j(T_g x) \, d\mu(g) = \Phi_j(x)$. Hoeffding's inequality gives, for any $t > 0$,

$$\mathbb{P}\left( \left| \frac{1}{K} \sum_{k=1}^{K} Z_{k,j} - \Phi_j(x) \right| \geq t \right) \leq 2 \exp\left( -\frac{Kt^2}{2B^2} \right). \tag{A.6}$$

Applying a union bound over $j$ yields

$$\mathbb{P}\left( \|\widehat{\Phi}_K(x) - \Phi(x)\|_\infty \geq t \right) \leq 2d \exp\left( -\frac{Kt^2}{2B^2} \right). \tag{A.7}$$

Using $\|v\|_2 \leq \sqrt{d}\|v\|_\infty$ and setting $t = \varepsilon/\sqrt{d}$ proves the claim. $\square$

**Theorem A.3** (Approximate invariance of multi-shot mean pooling). *Assume the conditions of Lemma A.2 hold for both $x$ and $T_h x$ (i.e., $\sup_{g \in G} \|\phi(T_g x)\|_2 \leq B$ and $\sup_{g \in G} \|\phi(T_g T_h x)\|_2 \leq B$). If $g_1, \ldots, g_K \overset{i.i.d.}{\sim} \mu$, then for any $h \in G$ and any $\varepsilon > 0$, with probability at least $1 - 4d \exp\left( -\frac{K\varepsilon^2}{2B^2 d} \right)$,*

$$\left\| \widehat{\Phi}_K(T_h x) - \widehat{\Phi}_K(x) \right\|_2 \leq 2\varepsilon. \tag{A.8}$$

*Proof.* By Theorem A.1, $\Phi(T_h x) = \Phi(x)$ for any $h \in G$. Thus, by triangle inequality,

$$\|\widehat{\Phi}_K(T_h x) - \widehat{\Phi}_K(x)\|_2 \leq \|\widehat{\Phi}_K(T_h x) - \Phi(T_h x)\|_2 + \|\widehat{\Phi}_K(x) - \Phi(x)\|_2. \tag{A.9}$$

Apply Lemma A.2 to $x$ and to $T_h x$, then take a union bound. $\square$

**Deterministic canonical views matching the implementation.** In our implementation, the shots $\{g_k\}_{k=1}^{K}$ are fixed canonical views, which correspond to a deterministic quadrature rule. Define the empirical measure $\mu_K := \frac{1}{K}\sum_{k=1}^{K}\delta_{g_k}$, so that $\widehat{\Phi}_K(x) = \int_G \phi(T_g x)\, d\mu_K(g)$. Then the approximate invariance follows from controlling the quadrature error.

**Lemma A.4** (Deterministic quadrature error via discrepancy). *Fix $x \in \mathcal{X}$ and assume $g \mapsto \phi(T_g x)$ is $L$-Lipschitz on $G$ under a bi-invariant metric $d_G$: $\|\phi(T_g x) - \phi(T_{g'} x)\|_2 \le L\, d_G(g, g')$ for all $g, g' \in G$. Then*

$$\|\widehat{\Phi}_K(x) - \Phi(x)\|_2 \le L\, W_1(\mu_K, \mu), \tag{A.10}$$

*where $W_1$ is the 1-Wasserstein distance induced by $d_G$.*

**Theorem A.5** (Approximate invariance of multi-shot pooling). *Under the assumptions of Lemma A.4 for both $x$ and $T_h x$, for any $h \in G$,*

$$\|\widehat{\Phi}_K(T_h x) - \widehat{\Phi}_K(x)\|_2 \le 2L\, W_1(\mu_K, \mu). \tag{A.11}$$

*Proof.* Using Theorem A.1 and the triangle inequality as in Theorem A.3,

$$\|\widehat{\Phi}_K(T_h x) - \widehat{\Phi}_K(x)\|_2 \le \|\widehat{\Phi}_K(T_h x) - \Phi(x)\|_2 + \|\widehat{\Phi}_K(x) - \Phi(x)\|_2.$$

Apply Lemma A.4 to $x$ and $T_h x$. $\qquad\square$

# B. Supplementary Experiment Setup

## B.1. Dataset details

**EDBench. (Xiang et al., 2025)** To overcome the limitations of existing MLFFs that ignore electron density due to data scarcity, EDBench was introduced as a large-scale, high-fidelity dataset. It is built upon the PCQM4Mv2 (Nakata & Shimazaki, 2017) dataset and provides accurate DFT-computed electron densities for over 3.3 million molecules. EDBench moves beyond simple energy/force prediction by offering a suite of ED-centric tasks, including prediction, retrieval, and generation.

**HOPV15. (Lopez et al., 2016)** The Harvard Organic Photovoltaic Dataset (HOPV15) is a collation of experimental photovoltaic data from the literature, and corresponding quantum-chemical calculations performed over a range of conformers, each with quantum chemical results using a variety of density functionals and basis sets. The HOPV15 dataset contains 350 p-type small molecules and polymers curated from the literature. For each molecule, experimental properties and corresponding quantum-chemical data are provided in XYZ format. Up to 20 low-energy conformers per molecule are included, along with available photovoltaic metrics such as power conversion efficiency (PCE), open-circuit voltage ($V_{\text{OC}}$), short-circuit current density ($J_{\text{SC}}$), and fill factor (FF).

**OPEP2. (Greenstein & Hutchison, 2023)** A dataset of 1225 experimentally reported non-fullerene acceptor (NFA)/donor devices was curated from the literature, including photovoltaic parameters such as $V_{\text{OC}}$, $J_{\text{SC}}$, FF, and PCE, as well as optical and electronic properties when available. To reduce computational cost, side chains were truncated to ethyl groups. After filtering for unique donor-acceptor core pairs and selecting the device with the highest reported PCE per pair, the final dataset contained 1000 pairs (599 unique acceptors and 210 unique donors).

**Deep4Chem. (Joung et al., 2020)** Deep4Chem is an open, curated experimental database of organic chromophores that aggregates 20,236 chromophore-environment records spanning 7,016 unique chromophores measured in 365 solvents or solid-state matrices, distributed as CSV with SMILES provided for both chromophores and solvents. It reports key photophysical properties, including absorption and emission peak wavelengths and bandwidths, extinction coefficient, photoluminescence quantum yield, and fluorescence lifetime. The dataset was compiled from the literature with manual curation and explicit quality-control rules to filter outliers and ensure consistency.

## B.2. Hyperparameters of pre-training and fine-tuning

We describe the pre-training hyperparameters of VisionED in Table 8. We set a learning rate of 0.005, a batch size of 256, a momentum of 0.9, and a weight decay of 1e-4 for pre-training. To evaluate the effectiveness of the pre-training task, we split 10% of the data for validation and 10% of the data for testing. Finally, we declare the training platform. We pre-trained VisionED over 50 epochs on a server with Intel(R) Xeon(R) Platinum 8375C@2.90GHz CPU and 8 NVIDIA 4090 (24GB)

*Table 6.* Basic statistical information of the Downstream dataset. For each descriptor, the minimum, maximum, and average values are reported.

| Dataset | Task | Unit | Min | Max | Mean |
|---|---|---|---|---|---|
| OPEP2 | PCE | % | 0.01 | 18.77 | 9.5 |
| | $J_{SC}$ | $mA \cdot cm^{-2}$ | 0.08 | 28.66 | 16.58 |
| | FF | % | 23.74 | 81.1 | 63.39 |
| | $V_{OC}$ | V | 0.22 | 1.34 | 0.88 |
| HOPV15 | PCE | % | 0.0005 | 10.2 | 3.83 |
| | $J_{SC}$ | $mA \cdot cm^{-2}$ | 0.04 | 18.2 | 8.74 |
| | FF | % | 22 | 77 | 54.36 |
| | $V_{OC}$ | V | 0.16 | 1.51 | 0.74 |
| Deep4Chem | $\lambda_{Abs}$ | nm | 127.7 | 1055.02 | 427.62 |
| | $\lambda_{Emi}$ | nm | 247 | 1050 | 500.24 |
| | PLQY | - | 0 | 1 | 0.35 |
| | FWHM | nm | 0 | 253.5 | 74.41 |

GPUs, which took about 4 days. We appended a MLP after the encoder for fine-tuning downstream tasks. In detail, we use the pre-trained encoder as the backbone to extract the features of molecular ED images and input them into an MLP (dropout1→linear1→activator →dropout2→linear2) for prediction. Following previous work, we use grid search to find the best combination of hyperparameters on downstream tasks. The specific search spaces are shown in Table 8. We run all fine-tuning experiments on a single NVIDIA 4090 GPU.

### B.3. Baselines

We summarize the detailed settings of all baselines in Table 7. For the ML baselines (SVM (Zhang, 2001), RF (Breiman, 2001)), we generated ECFP descriptors (Rogers & Hahn, 2010) as specified and conducted hyperparameter selection exclusively on the validation set; the final model, using the validation-selected configuration, was then evaluated once on the held-out test set, with the test set never used for model selection. GeoFormer (Wang et al., 2023) and EquiformerV2 (Liao et al., 2024) are Transformer-based architectures that use Interatomic Positional Encoding (IPE) and higher-degree tensors, respectively, to learn the interaction relationships between atoms. Unlike GeoFormer and EquiformerV2, which are specifically designed for molecules, PointVector (Deng et al., 2023) and X-3D (Sun et al., 2024b) are the latest methods that focus on real-world point clouds. ImageMol (Zeng et al., 2022) is an image-based molecular property-prediction approach that treats molecular 2D structures as pixel images to learn chemical representations for downstream tasks. By contrast, VideoMol (Xiang et al., 2024) is a molecular video-derived model that renders each molecule's 3D conformer as a 60-frame dynamic video to learn representations for property prediction. To ensure a fair comparison, we followed the official configurations for the deep learning models.

## C. Supplementary Method

### C.1. Density functional theory

DFT (Bartolotti & Flurchick, 1996; Medvedev et al., 2017) reformulates the many-electron ground-state problem in terms of the electron density $\rho(\mathbf{r})$ rather than the many-body wavefunction. The Hohenberg-Kohn theorems show that (i) the ground-state energy is a unique functional of $\rho$ and (ii) the exact ground-state density minimizes this energy functional. In practice, DFT is realized via the Kohn-Sham (KS) scheme, which maps the interacting system to noninteracting electrons moving in an effective potential, leading to one-electron KS equations:

$$\left[ -\frac{1}{2}\nabla^2 + V_{\text{eff}}(\mathbf{r}) \right] \psi_i(\mathbf{r}) = \varepsilon_i \, \psi_i(\mathbf{r}), \tag{C.1}$$

where $\psi_i(\mathbf{r})$ and $\varepsilon_i$ are, respectively, the wavefunction and energy of the $i$-th single-electron orbital. $V_{\text{eff}}(\mathbf{r})$ is the effective single-electron potential energy, defined as

$$V_{\text{eff}}(\mathbf{r}) = V_{\text{ext}}(\mathbf{r}) + V_{\text{H}}(\mathbf{r}) + V_{\text{xc}}(\mathbf{r}). \tag{C.2}$$

*Table 7.* Summary of baseline methods and VisionED.

| Category | Method | Pretrain | Data | Representation | Downstream | Hyperparameters | Procedure |
|---|---|---|---|---|---|---|---|
| Traditional ML Models | SVM | ✗ | – | ECFP | Trained from scratch | Hyperparameter search | Standard sklearn |
| | RF | ✗ | – | ECFP | Trained from scratch | Hyperparameter search | Standard sklearn |
| Point Cloud Models | PointVector | ✗ | – | ED point cloud | Trained from scratch | Followed by original paper | Original |
| | X-3D | ✗ | – | ED point cloud | Trained from scratch | Followed by original paper | Original |
| Geometric Graph Models | GeoFormer | ✗ | – | Geometric graph | Trained from scratch | Followed by original paper | Original |
| | EquiformerV2 | ✓ | S2EF-2M | Geometric graph | Initialized from pretrained weights | Followed by original paper | Original |
| Visual Molecular Models | ImageMol | ✓ | PubChem-10M | Molecular image | Initialized from pretrained weights | Followed by original paper | Original |
| | VideoMol | ✓ | PCQM4Mv2-2M | Molecular video | Initialized from pretrained weights | Followed by original paper | Original |
| | VisionED | ✓ | EDBench-2M | Multi-shot ED images | Initialized from pretrained weights | – | – |

The external potential $V_{\text{ext}}(\mathbf{r})$ is typically provided by the atomic nuclei. $V_{\text{H}}(\mathbf{r})$ is the Hartree potential, which is represented by the convolution of the electron density with the Coulomb kernel. The exchange-correlation potential $V_{\text{xc}}(\mathbf{r})$ is the variational derivative of the exchange-correlation energy functional.

The solution depends on the electron density and an exchange-correlation (XC) functional, for which various approximations exist (e.g., LDA (Becke, 1988), GGA (Perdew et al., 1996), and hybrid functionals). In this paper, the exchange-correlation functional used is B3LYP, and the 6-31G\*\*/+G\*\* basis set is selected. B3LYP integrates the advantages of the Hartree-Fock method and DFT. The 6-31G\*\*/+G\*\* basis set enhances computational accuracy by splitting the valence electron orbitals into two sets of basis functions and further incorporating diffuse functions. This combination achieves a good balance between precision and efficiency, making it more suitable.

### C.2. ED image generation pipeline

To clearly describe the ED image generation pipeline, we present the overall process in Algorithm 1, which comprises three stages. First, molecular geometry files are converted into Psi4-compatible inputs (Turney et al., 2012; Smith et al., 2020), and single-point DFT calculations are performed to generate corresponding ED and electrostatic potential (ESP) cube files. In the second stage, PyMOL (DeLano et al., 2002) is used to load the molecular structures and cube files, apply an ESP-based color ramp, and construct isosurfaces or meshes from ED data at a specified isovalue, enabling clear visualization of electrostatic features. Finally, in the third stage, multi-view rendering is conducted by rotating the molecular system along predefined axes, and images are captured from multiple angles to produce high-quality, standardized ED visualization datasets suitable for downstream analysis and model training. Since the molecular geometries are already optimized at the quantum DFT level, we perform single-point calculations on these pre-optimized structures to obtain additional quantum chemical properties, such as electron density. This approach eliminates the need for full geometry optimization. Instead, each molecule undergoes just one SCF calculation, which is consistent with the time estimated for a single SCF cycle at the B3LYP/6-31G\*\*/+G\*\* level on a single CPU. Thus, the 205,000 core-hours reported correspond to the computational cost for these single-point SCF calculations.

During fine-tuning, molecules in downstream tasks do not contain corresponding conformational information, so we followed VideoMol (Xiang et al., 2024) to obtain molecular conformers through a multi-stage generation method. To generate 3D conformations, we first removed hydrogen atoms and applied MMFF94 optimization using RDKit with a maximum of 5000 iterations. If the optimization failed to converge, the iteration limit was doubled iteratively up to 10 times. If convergence was still not achieved, or RDKit failed to generate a conformer, we used the 2D structure instead. We use the semi-empirical GFN2-xTB (Bannwarth et al., 2019) method to refine conformers, as it offers a fast yet sufficiently accurate quantum-level

*Table 8.* The parameter details in the pre-training and fine-tuning phase.

| Model details | |
| --- | --- |
| Backbone of VisionED | 12-layer Vision Transformer with 16 patches |
| EGPClassifier | |
| QSPClassifier | |
| MaxTypeClassifier | |
| MinTypeClassifier | 2 layers of fully connected neural network |
| MaxMinValueClassifier | |
| FinetuneClassifier | |
| **Pre-training parameters** | |
| Learning Rate | 5e-3 |
| Batch Size | 256 |
| Momentum | 0.9 |
| Weight Decay | 1e-4 |
| Learning Rate Decay | Linear |
| Image Size | 6×3×224×224 |
| Training Epochs | 50 |
| $\lambda_1$ ,$\lambda_2$, and $\lambda_3$ | 1 |
| **Fine-tuning parameters** | |
| $K$ shot images | 6 (top, bottom, left, right, front, and back) |
| Learning Rate | [1e-5, 1e-4, 5e-4, 1e-3, 5e-3, 1e-2, 5e-2, 0.1] |
| Batch Size | [8, 16, 32, 64] |
| Momentum | 0.9 |
| Weight Decay | 1e-5 |
| Learning Rate Decay | Linear |
| Image Size | 6×3×224×224 |
| Training Epochs | [10, 25] |
| **Training Platform** | |
| CPU | Intel(R) Xeon(R) Platinum 8375C@2.90GHz |
| GPU | 8×NVIDIA 4090 (24 GB) |

approximation. The optimized structures are then used to generate multi-shot ED images, following the same rendering protocol used in pre-training to ensure consistency in the input representation.

### C.3. ESSOR descriptor generation

We utilized Multiwfn in batch mode to perform electron density and related wavefunction analysis based on the files obtained from DFT calculations. The input script automates the Multiwfn menu-driven interface and carries out the procedures (Algorithm 2). We obtain 19 electron descriptors, 11 ESP descriptors, 6 orbital descriptors, and 11 shape descriptors, for a total of 47 descriptors. Five energy descriptors (such as Nuclear Repulsion Energy, One-Electron Energy, Two-Electron Energy, DFT Exchange-Correlation Energy, and Total Energy), 8 orbital descriptors (HOMO-2, HOMO-1, E_HOMO, E_LUMO, LUMO+1, LUMO+2, LUMO+3, HOMO-LUMO gap), and 9 ESP descriptors (Dipole X1, Dipole X2, Dipole X3, Dipole Y1, Dipole Y2, Dipole Y3, Dipole Z1, Dipole Z2, and Dipole Z3) are directly extracted from the outputs of Psi4 during DFT calculation. Finally, we get 69 ESSOR descriptors.

## D. Supplementary Results

### D.1. Pre-training results

Figure 8 and Figure 9 collectively demonstrate the effectiveness of the VisionED pre-training scheme on multiple tasks. During pre-training (Figure 8a), the total loss decreases steadily, indicating stable convergence. The ESSOR-Guided

---

**Algorithm 1** The overall process of ED image generation

---

**Data:** XYZ molecular geometry files $\mathcal{F}_{xyz}$

**Stage I:** ED and ESP cube file generation by Psi4.

**for** sampled xyz file $\mathcal{M}_{xyz} \in \mathcal{F}_{xyz}$ **do**

    **#1** Convert RDKit $\mathcal{M}_{xyz}$ to a Psi4 molecule.

        $\text{Psi4Mol} \leftarrow \text{SetupPsi4Mol}(\mathcal{M}_{xyz})$

    **#2** Setup reference (RHF/UHF), DFT method, basis set, and cubeprop parameters (grid spacing, overage, etc.).

        $\text{Reference} \leftarrow \text{SetupReferenceWavefunction}(\text{Psi4Mol})$

        $(\text{Method}, \text{Basis}) \leftarrow \text{SetupMethodNameAndBasisSet}(\text{Psi4Mol})$

        Set Psi4 options for `cubeprop`.

    **#3** Run single-point energy calculation.

        $(E, \text{Wfn}) \leftarrow \text{Psi4.energy}(\text{Method}/\text{Basis}, \text{return\_wfn} = \text{True})$

    **#4** Generate ED and ESP cube files.

        Run `Psi4.cubeprop(Wfn)`.

**end for**

**Output:** ED cube files $\mathcal{F}_{ED}$, ESP cube files $\mathcal{F}_{ESP}$, and log files $\mathcal{F}_{log}$

**Stage II:** Loading ED and ESP information with PyMOL.

**for** sampled $(\mathcal{M}_{xyz}, \mathcal{M}_{ED}, \mathcal{M}_{ESP})$ from $(\mathcal{F}_{xyz}, \mathcal{F}_{ED}, \mathcal{F}_{ESP})$ **do**

    **#1** Load xyz, ED, and ESP files.

        `load` $\mathcal{M}_{xyz}$, `XYZ`; `load` $\mathcal{M}_{ED}$, `ED`; `load` $\mathcal{M}_{ESP}$, `ESP`

    **#2** Create a color ramp legend.

        `ramp:` `ESP` $\in [-0.08, 0.0, 0.08] \rightarrow$ [`red`, `white`, `blue`]

    **#3** Create an isosurface with an isovalue of 0.05.

        `isosurface surface, ED, 0.05`

    **#4** Set the color of the ED surface using the legend.

        `set surface_color, legend, surface`

**end for**

**Stage III:** Multi-shot rendering of ED images.

**Input:** save path $P$, image height $\mathcal{H}$, image width $W$, rotation axis $X$, rotation angle $A$

**for** sampled $(\mathcal{M}_{xyz}, \mathcal{M}_{ED}, \mathcal{M}_{ESP})$ **do**

    **#1** Rotate the shot by $A$ degrees around axis $X$.

        `turn` $X, A$

    **#2** Render and save the image to $P$ with width $W$ and height $\mathcal{H}$.

        `png` $P$, `width=`$W$, `height=`$\mathcal{H}$

**end for**

**Output:** Multi-shot ED images

---

Pre-training (EGP) task shows consistent improvements in both MAE and $R^2$ metrics across epochs for validation and test sets (Figure 8b), confirming the model's learning capacity for electronic descriptors. Quantum Semantic Population (QSP) also achieves high accuracy (>0.85) early in training (Figure 8c), suggesting strong atom-level representation learning. For Charge-aware Atom-level Pre-training (CAP), both classification (Figure 8d-e) and regression (Figure 8f) tasks maintain high agreement between validation and test sets, with classification accuracy exceeding 0.98 and MAE dropping below 0.02.

Figure 9 further quantifies model performance across five descriptor categories: Electronic, ESP, Shape, Orbital, and Energy. The model achieves low MAEs and high $R^2$ scores for most descriptors, with minimal differences between validation and test sets, indicating good generalization. Notably, descriptors such as molecular polarity index, HOMO-LUMO gap, and dipole magnitude are predicted with high fidelity. Together, these results highlight the robustness and effectiveness of the VisionED architecture in capturing rich molecular quantum property information through pre-training.

**Pre-training gain.** Table 9 compares VisionED against the ImageNet-pretrained ViT-base model across all three benchmarks. Specifically, compared with the ViT-base baseline, VisionED achieves average gains of 23.34% / 14.96% on OPEP2, 13.83% on HOPV15, and 19.51% / 31.86% on Deep4Chem.

---

**Algorithm 2** Workflow for computing ESSOR descriptors via Multiwfn

---

**Input:** Geometry file

**Step 1:** Enter main menu.

    Input: `100`

**Step 2:** Calculate atom-specific properties from geometry.

    Input: `21`

**Step 3:** Report whole-system size information.

    Input: `size`

    Computed: 8 shape descriptors (Farthest distance, vdW radius of Atom1, vdW radius of Atom2, Diameter of the system, Radius of the system, Length, Width, Height)

**Step 4:** Exit submodule.

    Input: `0`

**Step 5:** Compute fragment planarity and deviation span.

    Input: `MPP`

    Computed: 2 shape descriptors (Molecular planarity parameter, Span of deviation from plane)

**Step 6:** Exit submodule.

    Input: `n`, then `q`

**Step 7:** Compute molecular van der Waals volume.

    Input: `3`, then `9,0.001,1.7`

    Computed: 1 electron descriptor (Volume)

**Step 8:** Exit submodule.

    Input: `0,0,0`, then `0`

**Step 9:** Compute electric dipole/multipole moments and electronic spatial extent.

    Input: `300`, then `5`, then `0`

    Computed: 4 ESP descriptors (Electronic spatial extent, Dipole Magnitude, $|Q_2|$, $|Q_3|$)

**Step 10:** Orbital decomposition via Hirshfeld analysis.

    Input: `8, 8, 1, h-1, h, l, l+1`

    Computed: 6 Orbkit descriptors (ODI_HOMO_1, ODI_HOMO, ODI_LUMO, ODI_LUMO_Add1, ODI_Mean, ODI_var)

**Step 11:** Exit submodule.

    Input: `0`, then `-10`

**Step 12:** Quantitative analysis of molecular surface.

    Input: `12`, then `0`, then `-1`

    Computed: 18 electron descriptors (Estimated density, Minimal value, Maximal value, Overall surface area, Positive surface area, Negative surface area, Overall average value, Positive average value, Negative average value, Balance of charges, Product of $\sigma_{\text{tot}}^2$ and $\nu$, Internal charge separation, Molecular polarity index, Nonpolar surface area, Polar surface area, Overall skewness, Positive skewness, Negative skewness)

**Step 13:** Compute ALIE and LEA; compute sphericity.

    Input: `2, 2, 0, -1, 2, 4, 0`

    Computed: 6 ESP descriptors (Minimum ALIE, Maximum ALIE, ALIE Average, Minimum LEA, Maximum LEA, LEA Average) and 1 shape descriptor (Sphericity)

**Step 14:** Exit submodule.

    Input: `-1`, then `-1`

**Step 15:** Integrate a function in whole space.

    Input: `100`, then `4`, then `100`

    Computed: 1 ESP descriptor (Sphericity)

**Step 16:** Return and quit.

    Input: `0`, then `q`

**Output:** Computed descriptor sets (shape, electron, ESP, and Orbkit descriptors)

---

*Table 9.* Performance improvements on photovoltaic and emitter benchmarks.

| $\Delta$ | $V_{\mathrm{OC}}$ | $J_{\mathrm{SC}}$ | FF | PCE |
|---|---|---|---|---|
| OPEP2 acceptor | 16.67% | 35.85% | 12.79% | 28.06% |
| OPEP2 donor | 12.05% | 27.40% | 7.08% | 13.29% |
| HOPV15 | 12.15% | 18.10% | 12.30% | 12.76% |

| $\Delta$ | $\lambda_{\mathrm{Abs}}$ | $\lambda_{\mathrm{Emi}}$ | PLQY | FWHM |
|---|---|---|---|---|
| Deep4Chem emitter | 25.12% | 17.35% | 12.54% | 23.01% |
| Deep4Chem solvent | 46.17% | 41.43% | 18.11% | 21.74% |

## D.2. Ablation studies

**Pre-training task.** To study the impact of the pre-training strategies on VisionED, we train VisionED with different pre-training tasks, including w/o pre-training, CAP, EGP, QSP, and combinations (Table 10). We found all three tasks independently improve performance over the w/o pre-training baseline, with EGP contributing the most significant gains. Combining any two tasks consistently yields additional improvements, indicating their complementarity. Notably, the full multi-task pre-training model, VisionED, achieves the best overall results, reducing PCE MAE by 28.1% and $J_{\mathrm{SC}}$ MAE by 35.9%, confirming that VisionED benefits from jointly learning complementary features. These findings confirm that (i) each pre-training task is beneficial on its own, (ii) multi-task pretraining yields additional improvements over single-task pretraining, and (iii) the full VisionED pretraining strategy is the most effective.

*Table 10.* The ablation study of the pre-training task in VisionED on the OPEP2 dataset under the acceptor scaffold split.

| Pre-training task | | | Performance | | | |
|---|---|---|---|---|---|---|
| CAP | EGP | QSP | $V_{\mathrm{OC}}$ | $J_{\mathrm{SC}}$ | FF | PCE |
| ✗ | ✗ | ✗ | 0.090 | 5.484 | 8.891 | 3.157 |
| ✓ | ✗ | ✗ | 0.088 | 4.060 | 8.823 | 2.749 |
| ✗ | ✗ | ✓ | 0.086 | 3.728 | 8.624 | 2.485 |
| ✗ | ✓ | ✗ | 0.080 | 3.530 | 8.024 | 2.353 |
| ✓ | ✗ | ✓ | 0.083 | 3.696 | 8.360 | 2.445 |
| ✓ | ✓ | ✗ | 0.078 | 3.601 | 7.860 | 2.304 |
| ✗ | ✓ | ✓ | 0.077 | 3.540 | 7.922 | 2.289 |
| ✓ | ✓ | ✓ | **0.075** | **3.518** | **7.754** | **2.271** |

**Various signals.** We conducted an ablation study comparing S, ED, ED+ESP, S+ED, and S+ED+ESP on HOPV15, as shown in Table 11. The results show that ED alone performs similarly to S, while S+ED improves over both S and ED, indicating that ED contributes useful information beyond the molecular structure alone. Finally, S+ED+ESP achieves the best performance, supporting the view that ED geometry and ESP cues are complementary. From a physical standpoint, this complementarity is expected. While ED describes the spatial distribution of electrons, ESP directly reflects the molecular electrostatic environment.

**Image setting.** We added experiments on image size and render software, which are factors that may influence the image. The results, as Table 12 and 13, show that the image size and software achieved similar performance.

*Table 11.* Performance comparison of various signals on HOPV15.

| Model | $V_{\mathrm{OC}}$ | $J_{\mathrm{SC}}$ | FF | PCE |
|---|---|---|---|---|
| S | 0.120(0.009) | 2.958(0.273) | 8.718(0.220) | 1.562(0.058) |
| ED | 0.121(0.010) | 2.984(0.214) | 8.825(0.623) | 1.590(0.064) |
| ED+ESP | 0.112(0.009) | 2.909(0.214) | 8.544(0.773) | 1.540(0.060) |
| S+ED | 0.110(0.008) | 2.897(0.174) | 8.525(0.623) | 1.500(0.083) |
| S+ED+ESP | **0.107(0.008)** | **2.856(0.195)** | **8.442(0.510)** | **1.465(0.081)** |

*Table 12.* Performance comparison under different image sizes.

| Size | $224 \times 224$ | $512 \times 512$ | $1024 \times 1024$ |
|---|---|---|---|
| PCE | $1.465_{(0.081)}$ | $1.466_{(0.074)}$ | $1.467_{(0.085)}$ |

*Table 13.* Performance comparison under different rendering software.

| Render software | PyMol | VMD |
|---|---|---|
| PCE | $1.465_{(0.081)}$ | $1.468_{(0.049)}$ |

## D.3. Multi-shot images analysis

**Multi-shot analysis.** The native 3D ED representation is too large to store and use efficiently at scale. Under a 1 TB storage budget, it can accommodate only about 0.2M molecules, whereas our 6-shot image representation can support up to 10M molecules, corresponding to about a 50× improvement in storage efficiency (Table 14). Importantly, although more compact, it still preserves substantial ED information. The 6-shot setting is already close to the performance of more informative 60-shot and 180-shot representations. Therefore, the key advantage of our proposed representation makes large-scale learning on ED feasible.

*Table 14.* Performance and storage efficiency under different numbers of shots.

| #shot | 1 | 6 | 60 | 180 |
|---|---|---|---|---|
| PCE | $1.596_{(0.159)}$ | $1.465_{(0.081)}$ | $1.450_{(0.046)}$ | $1.450_{(0.015)}$ |
| Storage efficiency | 300× | 50× | 5× | 1.6× |

**Sampling strategies.** Table 15 compares three shot-sampling strategies for multi-shot ED images. Notably, the canonical *orthographic* strategy demonstrates superior performance over random sampling, achieving both lower error (MAE: 1.465 vs. 1.481). Given that the Fibonacci approach offers no meaningful performance improvement despite its more complex sampling logic, we select the canonical orthographic views for their simplicity and computational efficiency.

*Table 15.* Comparison of different shot sample strategies.

| Multi-shot strategy | Orthographic | Random | Fibonacci sphere |
|---|---|---|---|
| PCE | $\mathbf{1.465_{(0.081)}}$ | $1.481_{(0.100)}$ | $1.466_{(0.088)}$ |

## D.4. Pairwise t-test

Across the 16 tasks in Tables 1 and 2, we find that VisionED exceeds one pooled standard deviation over the best baseline on 12/16 metrics. In addition, the pairwise t-tests show that 12/16 gains are statistically significant in Table 16.

*Table 16.* Pairwise t-tests across the three seeds with best baselines.

| Split | $V_{OC}$ | $J_{SC}$ | FF | PCE |
|---|---|---|---|---|
| OPEP2 acceptor | 0.328 | 0.042 | 0.038 | 0.004 |
| OPEP2 donor | 0.663 | 0.653 | 0.779 | 0.048 |

| Split | $\lambda_{Abs}$ | $\lambda_{Emi}$ | PLQY | FWHM |
|---|---|---|---|---|
| Deep4Chem emitter | $9 \times 10^{-7}$ | 0.001 | 0.007 | 0.023 |
| Deep4Chem solvent | $9 \times 10^{-15}$ | $2 \times 10^{-6}$ | 0.047 | 0.039 |

## D.5. End-to-end inference cost

In our pipeline, the reported runtime includes the full process of structure generation, ED generation, multi-shot image rendering, and model inference (Table 17). Based on 100 randomly sampled candidate pairs from the Deep4Chem test

dataset, the total runtime is 11.7 min for 100 pairs. We note, however, that the dominant cost arises from ED generation (79%), while the downstream image rendering and model inference are comparatively lightweight (7% and 3%, respectively).

*Table 17.* Time statistics, time is computed using 100 pairs randomly sampled from the Deep4Chem test dataset.

|  | Structure generation | ED generation | Multi-shot images render | Model inference |
|---|---|---|---|---|
| Time/min | 1.3 | 9.2 | 0.8 | 0.4 |

In addition, the cost of obtaining ED is being rapidly reduced by recent progress in DL-based ED methods (Jørgensen & Bhowmik, 2020; Li et al., 2025a). Once such methods become more reliable, the overall end-to-end runtime can be substantially reduced. To further examine the trade-off between efficiency and accuracy, we added an exploratory experiment using ED generated by DeepDFT and GFN2-xTB, as summarized in Table 18. These preliminary results suggest that faster approximate ED may improve the practical appeal of our framework.

*Table 18.* Comparison of DeepDFT and GFN2-xTB on the HOPV15 dataset. Total time refers to time of the ED generation for HOPV15.

| Model | DeepDFT | GFN2-xTB |
|---|---|---|
| PCE | 1.702$_{(0.098)}$ | 1.590$_{(0.064)}$ |
| Total time/min | 5.3 | 20.5 |

### D.6. High-throughput screening of promising D-A combinations

To accelerate the discovery of high-efficiency OPV materials, we employed VisionED to conduct large-scale virtual screening across a vast donor-acceptor chemical space. As shown in Figure 10a, a total of 210 donor and 599 acceptor molecules were combinatorially paired, generating 125,790 unique D-A combinations. Among these, only ∼1,000 combinations have been experimentally investigated, leaving the vast majority unexplored. VisionED was then applied to predict the PCE of each candidate in a high-throughput, automated manner. Using a PCE threshold of 15% to define high-performance pairs, 670 promising D-A combinations were identified for further analysis, involving 32 D structures and 62 A structures.

To extract representative and experimentally tractable structural units from this subset, we ranked all donors and acceptors by their frequency of appearance among the 670 high-PCE pairs. Structures in the top 10% of this ranking were selected as key components, resulting in 21 donors and 30 acceptors (with only the top 5% of acceptors shown in Figure 10b due to space constraints). The predicted PCEs of all combinations between these key structures are visualized in a heatmap (Figure 10b), where each axis corresponds to a specific donor or acceptor, and the color gradient from blue to red reflects increasing predicted efficiency. It is noted that D18 and PBDB-TF donors have been widely studied (Chen et al., 2024; Ma et al., 2024). Furthermore, this analysis revealed several high-potential combinations not yet studied in experiments. Many of these outperform well-known donor systems such as D18, including D18:BTP-PhC6 (17.94%), D18:H2 (17.85%), and D18:BTP-ClBr (17.76%), highlighted by yellow rectangles on the heatmap. **Remarkably, we also identified a donor-acceptor pair, D18:AQx-2 (Jiang et al., 2023), that was not included in the training data but has been experimentally reported in literature.** VisionED predicted a PCE of 17.15% for this combination, closely matching the experimental value of 17.20%, corresponding to a prediction accuracy of 99.70%. These results demonstrate VisionED's capacity to uncover novel, high-efficiency D-A pairs from an expansive candidate pool with minimal experimental data.

To further investigate the compatibility landscape between donors and acceptors, statistical methods were employed to quantify the contributions of donors and acceptors to the photovoltaic performance. Based on the predicted PCE values of 125,790 candidate D-A combinations, the contribution score of a given donor or acceptor to PCE was calculated. For example, the given donor D18 can be combined with various acceptors to generate 599 D-A combinations. Its contribution score is defined as the proportion of combinations whose PCE predicted value is above 15%. For the 210 donors and 599 acceptors in the present work, the contribution scores are calculated and normalized for structural analysis.

Figure 10c-d shows the top contribution scores of several representative structures of D and A. Among donors, D18 stands out with a normalized score of 1.0, followed by high-performing structures such as PBDB-TF (0.75), PBDB-T-2Cl (0.71), and PBDB-T-SF (0.71), all of which are widely used in state-of-the-art OPV devices. These results underscore the consistent value of PBDB-based donors across various acceptor partners. On the acceptor side, H2 (1.0) exhibits the strongest compatibility across donors, along with BTP-S2, BTP-ClBr, BTTPC-Br (all 0.94), and BTP-PhC6 (0.78). Except for BTP-S2, which adopts an asymmetric end-group design, the other top-scoring acceptors feature symmetric terminal

groups, highlighting that both symmetric and asymmetric architectures can support high efficiency, depending on the specific donor pairing and packing behavior. These results are consistent with experimental observations of these molecules in leading OPV systems (He et al., 2022). These insights offer data-driven guidance for the rational design of donor-acceptor combinations and demonstrate the practical utility of VisionED in accelerating materials discovery.

In summary, these results demonstrate that VisionED identifies multiple high-efficiency D-A pairs, including both novel predictions and literature-reported combinations, highlighting its practical screening potential. Through large-scale statistical analysis of combinations, VisionED quantifies the intrinsic contribution of individual donors and acceptors to photovoltaic performance, revealing high-performing structures, aligning well with experimental trends.

## E. Future Works

**Scalable ED image generation via surrogate modeling.** While ED images offer rich electronic insights, their reliance on DFT calculations creates a computational bottleneck for massive chemical libraries. To address this, we propose two strategies: (i) developing neural structure-to-ED surrogates to enable rapid, inference-only generation of ED images, bypassing expensive DFT steps for high-throughput screening; (ii) employing knowledge distillation to transfer electronic-level representations from VisionED to lighter, topology-based models. This ensures that the physical insights of ED are retained even when using computationally efficient architectures.

**Spatiotemporal ED dynamics.** Current ED images used by VisionED are static representations of equilibrium structures, while many key molecular processes, including geometry relaxation, charge transfer, and excited-state dynamics, are inherently time-dependent. Capturing the evolution of ED across these processes could provide a deeper mechanistic understanding and remains an open challenge. One promising approach is to utilize time-resolved ED trajectories derived from DFT calculations under structural optimization, vibrational analysis, or reaction pathways. By modeling the temporal evolution of electron density, we can construct dynamic ED sequences and train VisionED-like architectures using video-based temporal learning frameworks, enabling the model to learn from both spatial and temporal electronic patterns. This could open new avenues for understanding reactivity, excited-state dynamics, and non-equilibrium processes in complex molecular systems.

**Generalization and experimental validation.** Our current evaluation focuses on photovoltaics, leaving vast regions of the materials space unexplored. We aim to generalize VisionED to a broader spectrum of functional organic materials, including fluorescent probes and electrolytes, while expanding target properties to cover optical, thermodynamic, and device-level metrics. Furthermore, we plan to establish a closed-loop discovery workflow, where model predictions are rigorously verified via targeted laboratory synthesis and characterization, bridging the gap between computational screening and experimental reality.

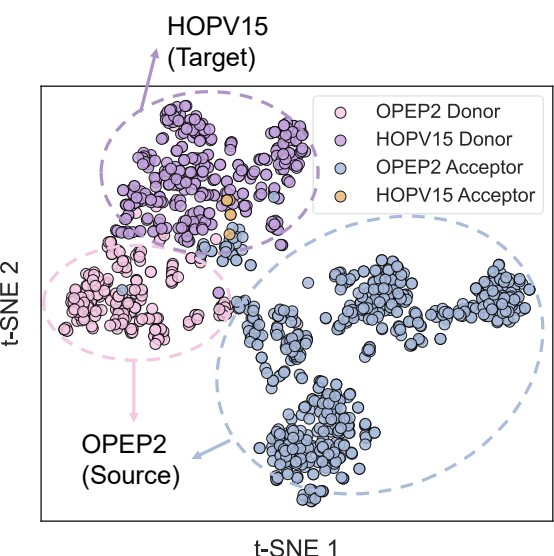

*Figure 7.* T-SNE visualization of molecular scaffolds from OPEP2 (donor/acceptor) and HOPV15 shows distinct structural separation.

**From screening to de novo design.** While VisionED excels as a discriminative model for screening existing scaffolds, it currently lacks generative capabilities. To transcend the boundaries of known chemical space, a critical direction is the development of an ED-conditioned generative framework. Such a model would aim to generate novel donor-acceptor architectures with targeted electronic distributions, enabling the de novo design of materials rather than mere selection, thus offering a new paradigm for functional material innovation.

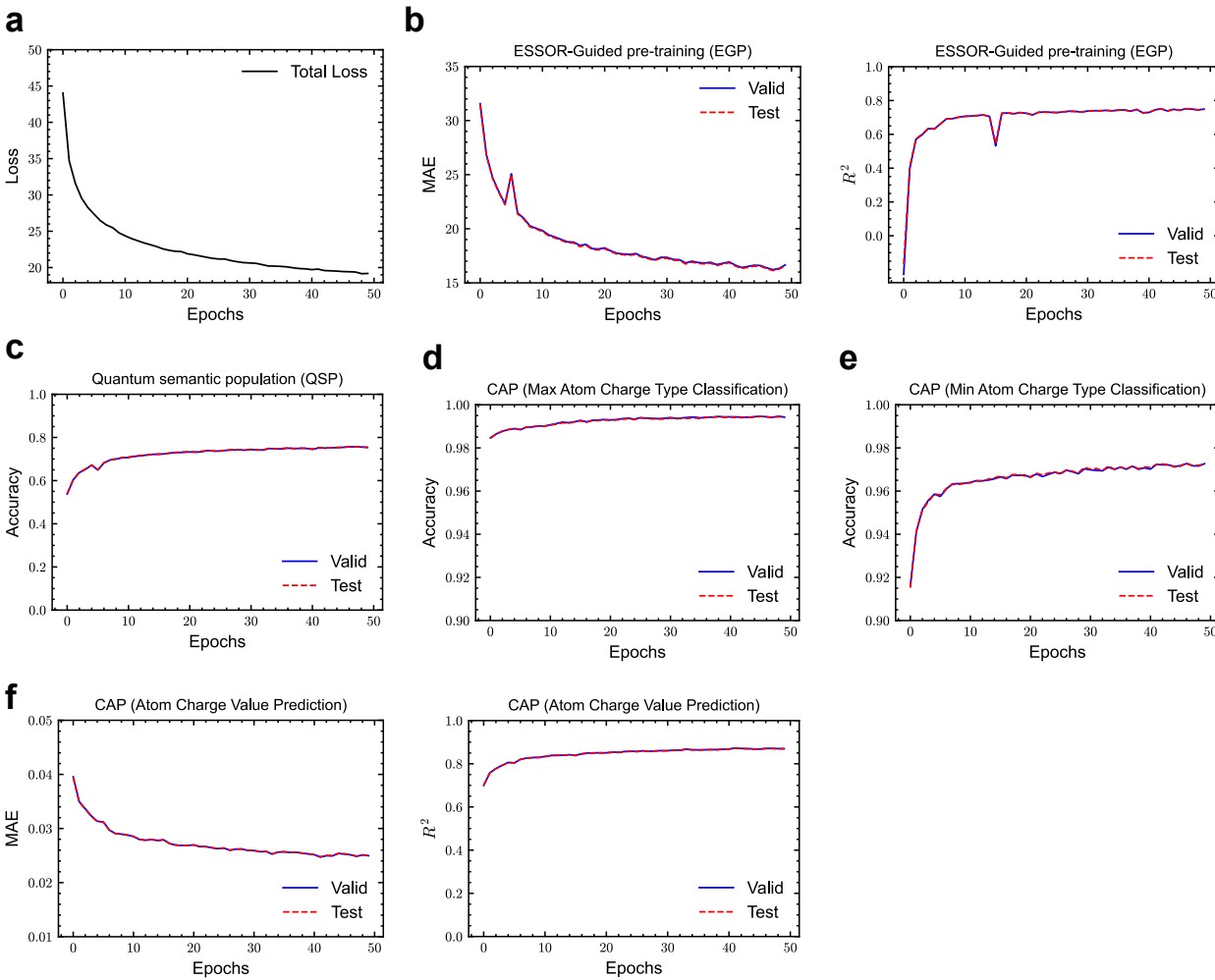

*Figure 8.* VisionED loss details and performance on validation/test set during the pre-training phase. The x-axis represents the training epoch, and the y-axis represents the corresponding metric, respectively. **(a)** Total loss across epochs during pre-training. **(b)** MAE and R2 for the ESSOR-Guided pre-training task (EGP). **(c)** Classification accuracy for quantum semantic population (QSP). **(d–e)** Accuracy for classification of the maximum and minimum atom charge types in molecules using CAP (Charge-aware Atom-level Pretraining). **(f)** MAE and R2 values for atom-level charge value regression using CAP.

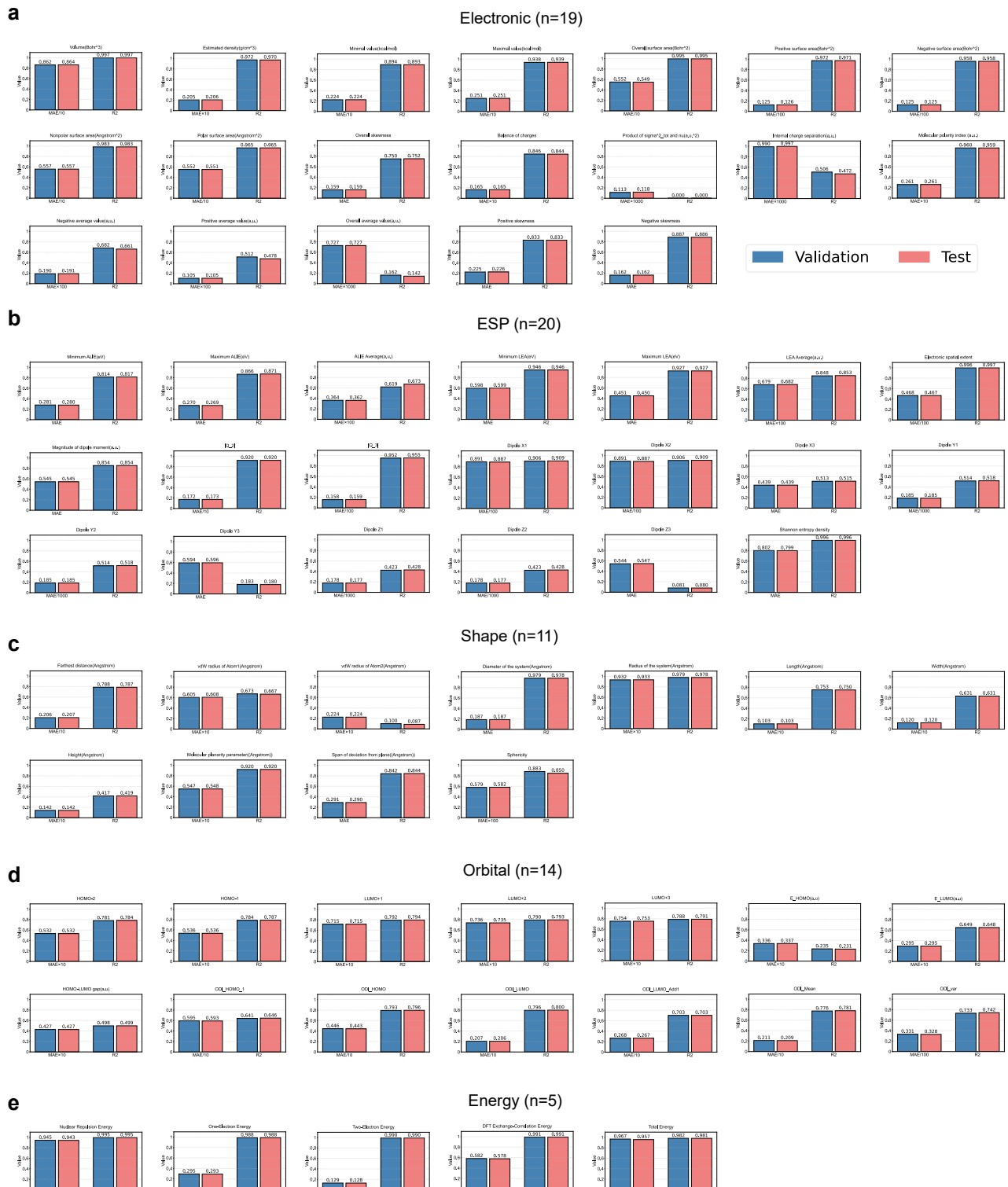

*Figure 9.* Validation and test performance of EGP. For each descriptor, blue and red bars represent results on the validation and test sets, respectively. MAE and $R^2$ values are displayed directly above the bars. For uniform display, we appropriately scale the MAE values.

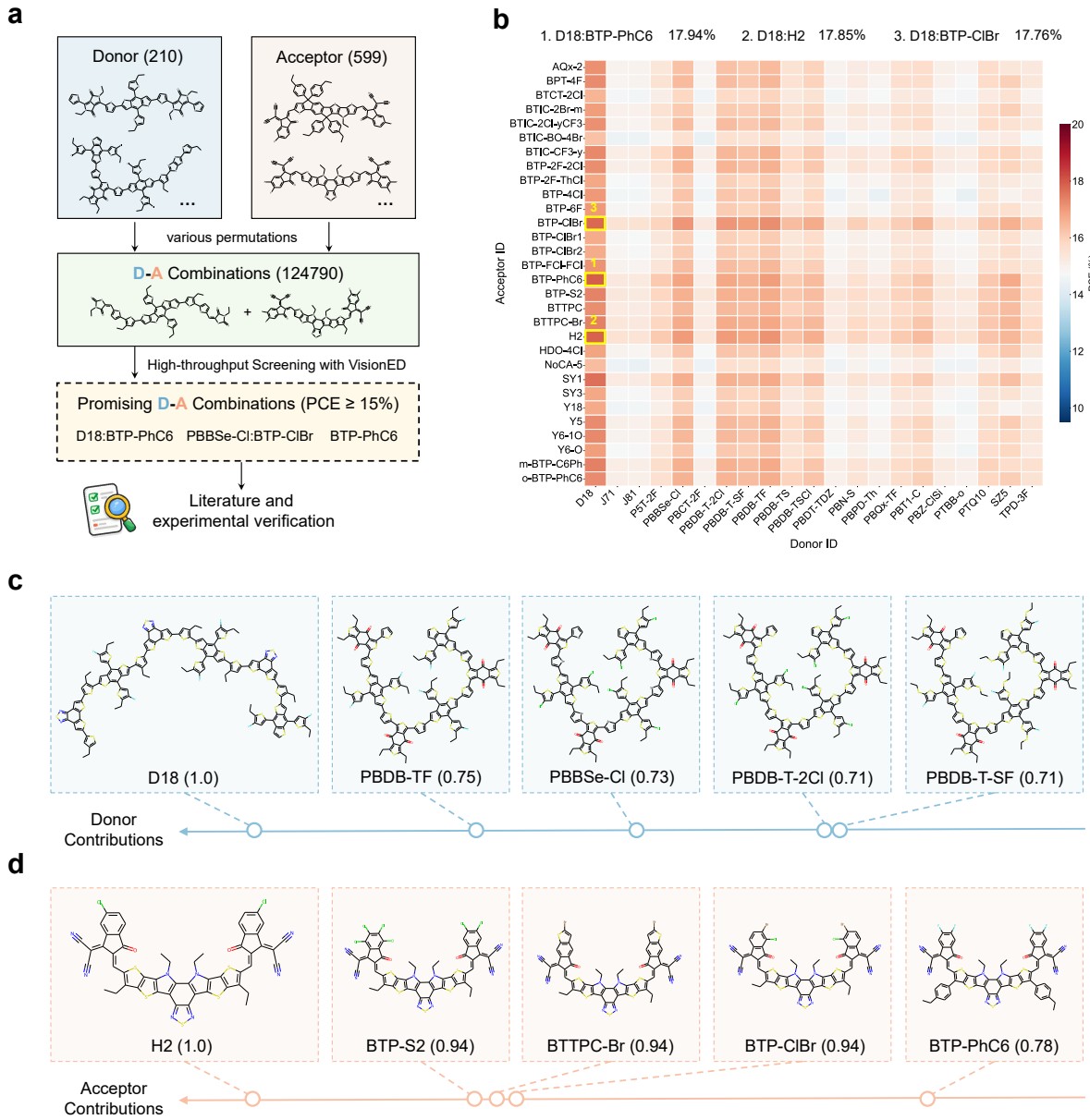

*Figure 10.* High-throughput virtual screening of promising donor-acceptor combinations using VisionED. **(a)**, Procedures for the screening of the promising D-A combinations with VisionED. **(b)**, Heatmap of the predicted PCE values for the promising combinations of key donor and acceptor, where the top-3 representative D-A combinations with high PCE are indicated by the yellow rectangles, and the corresponding predicted PCE values are given at the top of the heatmap. c-d, Top-5 representative donors **(c)** and acceptors **(d)**, along with their contribution scores indicating the relative importance of each component in high efficiency D-A combinations.

