# OpenReview forum: "Physics-Informed Pre-training on Efficient Electron-Density Images for Organic Material Property Prediction"
_ICML.cc/2026/Conference — ICML 2026 regular_

### Official Review · Reviewer_hcXc · 2026-02-27

**Soundness:** 2
**Presentation:** 3
**Significance:** 2
**Originality:** 3
**Overall Recommendation:** 4
**Confidence:** 4

**Summary:**

This paper proposes VisionED, which leverages electron density images to enhance the prediction of material properties. Electron density is the fundamental physical quantity that determines material properties, but electron density data is scarce, and its computation and storage costs are high. The authors constructed a dataset containing 12 million multi-shot electron density images, defined a descriptor set named ESSOR and designed cross-scale pre-training objectives, and subsequently fine-tuned the model on downstream tasks. The authors validated the approach through experiments on photovoltaic and organic chromophore datasets, achieving performance surpasses various baseline models.

**Compliance With Llm Reviewing Policy:**

Affirmed.

**Final Justification:**

My main concerns about the effectiveness of the proposed method were addressed by the additional experimental results provided in the rebuttal.

**Key Questions For Authors:**

Table 5 compares the performance of 1, 6, and 60-shot images taken along each of the three orthogonal axes (x, y, and z). What about using full electron density information, or using images taken along other directions?

**Limitations:**

yes

**Strengths And Weaknesses:**

Strengths:
- The proposed pretraining tasks guide the model to learn underlying physical laws across multiple scales, yielding plausible performance improvements.
- Multi-shot ED Images Learning strategy is proposed to reduce storage and computational costs.
- The experimental setup is comprehensive.

Weaknesses:
- I don't think data scarcity is a good motivation.  Any DFT calculation must first obtain the electron density. It's just that electron density is too large to be stored.
- Projecting the continuous 3D electron density field into a finite number of 2D images inevitably leads to information loss. Many factors can influence the resulting image representation.
- The correlation between the benchmark tasks and electron density is unclear. For instance, many properties of photovoltaic materials involve electron-phonon coupling or excited-state calculations, which cannot be captured by conventional ground-state DFT.
- Although the overall performance has improved, the gains compared to methods that do not rely on electron density (such as EquiformerV2) are not substantial enough. Considering the significant computational cost associated with electronic structure calculations, the new method is unlikely to serve as a highly compelling alternative.

---

> ### Author Rebuttal · Authors · 2026-03-30
>
> We appreciate your thorough review and helpful comments. We have carefully addressed your concerns below.
>
> > Q1: Electron density is too large to be stored.
>
> A1: A1: The native 3D ED representation is too large to store and use efficiently at scale. Under a 1 TB storage budget, it can accommodate only about 0.2M molecules, whereas our 6-shot image representation can support up to 10M molecules, corresponding to about a 50× improvement in storage efficiency (Table R9). Importantly, although more compact, it still preserves substantial ED information. The 6-shot setting is already close to the performance of more informative 60-shot and 180-shot representations. Therefore, the key advantage of our proposed representation makes large-scale learning on ED feasible.
>
> **Table R9**
>
> |#shot|1|6|60|180|
> |-|-:|-:|-:|-:|
> |PCE|1.596±0.159|1.465±0.081|1.450±0.046|1.450±0.015|
> |Storage efficiency|300×|50×|5×|1.7×|
>
> > Q2: Projecting the 3D into 2D images leads to information loss.
>
> A2.1: We clarify that our goal is not a lossless reconstruction, but an effective, scalable representation that promotes ED as an effective modality for material property prediction. Importantly, our method uses multiple shots to mitigate the information loss of rendering and recover a more holistic 3D electronic structure representation.  Table R9 shows that as the number of shots increases from 1 to 60, indicating that performance improves with more samples but saturates after 60 shots. This suggests that a limited number of shots is already sufficient to capture ED information.
>
> A2.2: We added experiments on image size and render software, which are factors that may influence the image. The results, as Table R10-11, show that the image size and software achieved similar performance.
>
> **Table R10**
>
> |#size|224*224|512*512|1024*1024|
> |-|-:|-:|-:|
> |PCE|1.465±0.081|1.466±0.074|1.467±0.085|
>
> **Table R11**
>
> |Render software|PyMol|VMD|
> |-|-:|-:|
> |PCE|1.465±0.081|1.468±0.049|
>
> > Q3: The correlation between the benchmark tasks and electron density is unclear.
>
> A3: Our claim is not that ground-state ED fully determines all properties, but provides a physically meaningful prior. As shown in Table R12, Structure+ED consistently outperforms either modality alone, indicating that ED contributes complementary task-relevant information beyond structural features. Moreover, VisionED relies on large-scale cross-scale physics pretraining, which can provide physics-informed inductive biases (eg, HOMO, 10.1126/science.ads0901)  beyond ground-state ED for downstream tasks. VisionED yields an average improvement of 22.7% over its counterpart without pretraining, further supporting this.
>
> **Table R12**
>
> |Model|Voc|Jsc|FF|PCE|
> |-|-:|-:|-:|-:|
> |Structure|0.120±0.009|2.958±0.273|8.718±0.220|1.562±0.058|
> |ED|0.112±0.009|2.909±0.214|8.544±0.773|1.540±0.060|
> |Structure+ED|0.107±0.008|2.856±0.195|8.442±0.510|1.465±0.081|
>
> > Q4: Significant computational cost associated with electronic structure calculations
>
> A4: Compared with EquiformerV2, the pairwise t-tests show that 12/16 gains are statistically significant in Table R13. In particular, the average relative improvement is 7.85% across three benchmarks.   In addition, the cost of obtaining ED is being rapidly reduced by recent progress in DL-based ED methods (DOI: 10.1038/s41524-022-00863-y; 10.1038/s41467-025-60095-8).  Once such methods become more reliable, the overall end-to-end runtime can be substantially reduced. To further examine the trade-off between efficiency and accuracy, we added an exploratory experiment using ED generated by DeepDFT and GFN2-xTB, as summarized in Table R1. These preliminary results suggest that faster approximate ED may improve the practical appeal of our framework.
>
> **Table R13**
>
> |p-value|Voc|Jsc|FF|PCE|
> |-|-:|-:|-:|-:|
> |OPEP2 acceptor|0.328|0.042|0.025|0.003|
> |OPEP2 donor|0.663|0.571|0.751|0.030|
> |p-value|$\lambda_{Abs}$|$\lambda_{Emi}$|PLQY|FWHM|
> |Deep4Chem emitter|9e-7|9e-5|0.002|0.015|
> |Deep4Chem solvent|1e-16|2e-6|0.047|0.039|
>
> > Q5: What about using full electron density information, or using images taken along other directions?
>
> A5: We have already included an analysis of view-selection strategies in Appendix Sec. D.3 and Table R14. Notably, the canonical orthographic strategy demonstrates superior performance over random sampling, achieving both lower error. Given that the Fibonacci approach offers negligible performance gains despite its more complex sampling logic, we select the canonical orthographic views for their simplicity and computational efficiency.
>
> **Table R14**
>
> |Multi-shotstrategy|Orthographic|Random|Fibonacci sphere|
> |-|-:|-:|-:|
> |PCE|1.465±0.081|1.481±0.100|1.466±0.088|
>
> We thank the reviewer for the constructive feedback, which has significantly improved our work. If our responses have addressed your concerns, we kindly ask you to consider raising your score. We remain open to further discussion if any points require additional clarification.

---

> > ### Author Rebuttal · Reviewer_hcXc · 2026-04-02
> >
> > I appreciate the author's clarification and the additional experimental results. I will raise my score.

---

> > > ### Author Response · Authors · 2026-04-05
> > >
> > > We sincerely thank the reviewer for the positive feedback and for recognizing our clarifications and additional experiments. We greatly appreciate your willingness to raise the score.

---

### Official Review · Reviewer_fxL8 · 2026-03-12

**Soundness:** 4
**Presentation:** 3
**Significance:** 4
**Originality:** 3
**Overall Recommendation:** 5
**Confidence:** 3

**Summary:**

This paper introduces VisionED, a physics-informed framework for organic material property prediction based on efficient multi-shot electron-density (ED) images rendered from DFT calculations. Instead of using high-dimensional ED point clouds, the paper represents each molecule with a small set of canonical ED views and encodes them with a shared vision backbone followed by view aggregation. On top of this representation, the authors pretrain the model using three auxiliary objectives tied to electronic structure, and then fine-tune it on downstream organic-material prediction tasks, including photovoltaic and chromophore-related benchmarks. The paper also studies low-data and out-of-distribution settings, compares the ED-image representation against prior ED point-cloud inputs, and demonstrates an application to donor–acceptor virtual screening.

**Compliance With Llm Reviewing Policy:**

Affirmed.

**Final Justification:**

My final recommendation remains Accept. I find the paper technically strong, significant, and clearly presented, with a genuinely useful contribution in bringing efficient ED-image pretraining into organic material property prediction. The method is well motivated, the empirical evaluation is broad, and the results suggest real practical value for low-data and experimentally relevant materials tasks.

The rebuttal addressed my main concerns well. In particular, the added ablation disentangling structure, ED, and ESP signals strengthens the claim that ED and ESP provide complementary information; the overlap check reduces concern about leakage from pretraining to downstream benchmarks; and the end-to-end runtime breakdown usefully clarifies the real deployment cost. I also appreciate the authors’ willingness to calibrate both the novelty framing and the use of the term “self-supervised.”

Overall, the rebuttal reinforces rather than changes my prior assessment. Some limitations remain, especially the current dependence on relatively expensive ED generation, but I view these as scope and practicality issues rather than flaws in soundness. For these reasons, I am keeping my original Accept recommendation.

**Key Questions For Authors:**

1. Since the rendering appears to combine ED geometry with ESP-based coloring, could you please provide ablations that disentangle these signals?

2. Is there any overlap or near-overlap between the large pretraining corpus and downstream benchmarks?

3, What is the estimated true end-to-end inference cost on unseen candidates?

**Limitations:**

Yes

**Strengths And Weaknesses:**

## Strengths

I find the core idea about how to incorporate more physically grounded information into molecular/material property prediction without making the representation prohibitively expensive compelling. Using ED-derived images as a compact surrogate for electronic structure is a reasonable and practically attractive design choice, and the multi-shot rendering strategy is simple enough to scale. The paper is also stronger than average empirically: it evaluates on multiple downstream datasets, includes low-data and cross-dataset settings, provides ablations over pretraining tasks and view strategies, and includes an efficiency comparison against a previous ED point-cloud representation.

The idea of pretraining on electronic-structure-derived modalities and transferring to scarce experimental materials tasks is important and timely. If robust, this could become a useful design pattern in AI for science: learn from a physics-rich but expensive modality once, then amortize that signal across downstream tasks.

I think the paper has real novelty, but the novelty should be framed more precisely. What seems genuinely new here is the combination of efficient ED-image rendering, large-scale pretraining on ED views, and downstream evaluation on experimentally relevant organic-material tasks. Relative to that literature, this paper’s novelty is best understood as bringing ED-image pretraining into organic material property prediction in an efficient way, rather than as the first broadly ED-aware or pretrained molecular model.

## Weaknesses

The author appears to entangle electron-density geometry with electrostatic-potential coloring, so it is unclear how much of the gain comes from ED itself versus ESP-like cues. An ablation with grayscale ED-only rendering, ESP-only rendering, or geometry-only surfaces would make the conclusions much better

The paper describes the approach as “self-supervised,” but two of the pretraining objectives are closer to automatic supervision from DFT-derived targets/pseudo-labels than to label-free self-supervision in the usual representation-learning sense. I do not view this as a big drawback, but I do think the term should be calibrated more carefully.

---

> ### Author Rebuttal · Authors · 2026-03-30
>
> We appreciate your thorough review and helpful comments. We have carefully addressed your concerns below and welcome any additional feedback.
>
> > Q1: Could you please provide ablations that disentangle these signals?
>
> A1: Following this suggestion, we conducted an ablation study comparing S, ED, ED+ESP, S+ED, and S+ED+ESP on HOPV15, as shown in Table R7. The results show that ED alone performs similarly to S, while S+ED improves over both S and ED, indicating that ED contributes useful information beyond the molecular structure alone. Finally, S+ED+ESP achieves the best performance, supporting the view that ED geometry and ESP cues are complementary. From a physical standpoint, this complementarity is expected. While ED describes the spatial distribution of electrons, ESP directly reflects the molecular electrostatic environment.
>
> **Table R7: Performance comparison of various signals on HOPV15.**
>
> |Model|Voc|Jsc|FF|PCE|
> |-|-:|-:|-:|-:|
> |S|0.120±0.009|2.958±0.273|8.718±0.220|1.562±0.058|
> |ED|0.121±0.010|2.984±0.214|8.825±0.623|1.590±0.064|
> |ED+ESP|0.112±0.009|2.909±0.214|8.544±0.773|1.540±0.060|
> |S+ED|0.110±0.008|2.897±0.174|8.525±0.623|1.500±0.083|
> |S+ED+ESP|**0.107±0.008**|**2.856±0.195**|**8.442±0.510**|**1.465±0.081**|
>
> > Q2: Is there any overlap or near-overlap between the large pretraining corpus and downstream benchmarks?
>
> A2: We explicitly checked for overlap between the large pretraining corpus and all downstream benchmarks using exact matching of canonical SMILES. Specifically, we canonicalized molecules in the pre-training set and in each downstream dataset, and ensured that no downstream molecule shares the same canonical SMILES with any molecule in the pre-training corpus. To further evaluate potential near-overlap in coarse molecular statistics, we compared atom count distributions. The pre-training corpus has an average atom count of 31, while HOPV15 and Deep4Chem have averages of 43 and 37, respectively. This suggests partial overlap in molecule size ranges. We will clarify these checks and statistics in the revised manuscript.
>
> > Q3: What is the estimated true end-to-end inference cost on unseen candidates?
>
> A3: In our pipeline, the reported runtime includes the full process of structure generation, ED generation, multi-shot image rendering, and model inference (Table R8). Based on 100 randomly sampled candidate pairs from the Deep4Chem test dataset, the total runtime is 11.7 min for 100 pairs. We note, however, that the dominant cost arises from ED generation (79%), while the downstream image rendering and model inference are comparatively lightweight (7% and 3%, respectively).  As also discussed in Appendix Sec. E, we view the development of a reliable and faster ED generation method as an important future direction. Once such a method becomes available, the overall end-to-end runtime can be substantially reduced.
>
> To further investigate this issue, we added an exploratory experiment comparing ED generated by DeepDFT (DOI: 10.1038/s41524-022-00863-y, an EGNN-based method for fast ED generation) and GFN2-xTB, as shown in Table R1. Although using DeepDFT-generated ED as input leads to slightly lower downstream performance than using higher-fidelity ED, it substantially reduces ED generation time. This preliminary result suggests a promising efficiency and accuracy trade-off.
>
> **Table R8: Time statistics, time is computed using 100 pairs randomly sampled from the Deep4Chem test dataset.**
>
> |  | Structure generation | ED generation | Multi-shot images render | Model inference |
> |---|---:|---:|---:|---:|
> | Time / min | 1.3 | 9.2 | 0.8 | 0.4 |
>
> **Table R1**
>
> | Model          | DeepDFT | GFN2-xTB |
> |----------------|---:|---:|
> | PCE            | 1.702 ± 0.098 | 1.590 ± 0.064 |
> | Total time/min | 5.3 | 20.5 |
>
> > Q4: Other Comments Or Suggestions
>
> A4.1: We agree that, relative to the existing literature, the novelty of our work is more accurately characterized as introducing ED-image pre-training into organic material property prediction in an efficient manner. We appreciate the reviewer’s suggestion to calibrate this point more carefully. In the revised manuscript, we will accordingly refine our wording to describe the novelty and contribution of this work more precisely.
>
> A4.2: We agree that describing the approach as “self-supervised” may be too broad. We appreciate this helpful suggestion and will calibrate our terminology more carefully in the revised manuscript. Specifically, we will avoid using “self-supervised” in an overly broad way and instead describe our method more precisely as a pre-training framework.
>
> We thank the reviewer for the constructive feedback, which has significantly improved our work. If our responses have addressed your concerns, we kindly ask you to consider raising your score. We remain open to further discussion if any points require additional clarification.

---

> > ### Author Rebuttal · Reviewer_fxL8 · 2026-04-06
> >
> > Thank you for the rebuttal. My concerns have been adequately addressed. The additional ablation separating structure, ED, and ESP signals was particularly helpful, and the clarification on pretraining/downstream overlap and full end-to-end inference cost also strengthened my confidence in the paper. I also appreciate the authors’ willingness to refine the novelty framing and terminology in the revision. Overall, the rebuttal reinforces my original positive assessment, so I am keeping my original score.

---

> > > ### Author Response · Authors · 2026-04-07
> > >
> > > Thank you very much for your thoughtful follow-up and for maintaining your positive assessment of our work.  We also greatly appreciate your constructive suggestions on refining the novelty framing and terminology, which will help us improve the final version. We will carefully incorporate all results and discussions into the revised version of the manuscript to ensure it meets the highest standards. Thank you again for your positive evaluation and encouraging feedback.

---

### Official Review · Reviewer_N64c · 2026-03-12

**Soundness:** 3
**Presentation:** 2
**Significance:** 2
**Originality:** 3
**Overall Recommendation:** 4
**Confidence:** 4

**Summary:**

VisionED represents electron density (ED) as six multi-view 224x224 images per molecule rather than costly volumetric point clouds, constructs a 2M-molecule DFT-computed corpus with 69-dimensional ESSOR quantum descriptors, and pre-trains a ViT-Base encoder via three objectives: atomic charge prediction (CAP), ESSOR regression (EGP), and quantum semantic clustering (QSP). The model is evaluated on OPEP2, HOPV15, and Deep4Chem benchmarks under scaffold splits, OOD transfer, and low-data regimes, where it is compared against geometric graph models, point cloud methods, and image-based baselines.

**Compliance With Llm Reviewing Policy:**

Affirmed.

**Key Questions For Authors:**

1. Table 8 compares against a randomly initialized ViT. What is the downstream performance of an ImageNet-pretrained ViT fine-tuned on OPEP2, HOPV15, and Deep4Chem? If ED-specific pre-training provides no additional lift over a standard pretrained backbone, the physics-informed motivation needs to be reframed. A satisfactory answer includes this control across all three benchmarks.

2.For how many metrics in Tables 1 and 3 does VisionED's gain over the best baseline exceed one pooled standard deviation? Pairwise t-tests across the three seeds would clarify which results are reliable. If most gains collapse within noise, the performance claims in Sections 5.2.1 and the abstract require revision.

3.The ternary blend evaluation in Table 4 covers five systems with no confidence intervals. Expanding to at least 15 systems and reporting variance on the accuracy metric would let this result stand on its own. As written, 93.7% accuracy on five points is not a defensible statistical claim.

4.Fine-tuning uses MMFF94 and GFN2-xTB conformers while pre-training uses B3LYP/6-31G** ED. Has the distributional shift between these two protocols been characterized, for example by comparing ESSOR descriptors for matched molecules under both? A satisfactory answer shows this gap is small enough not to undermine the core argument that ED images carry meaningful electronic information at inference time.

**Limitations:**

Partially addressed. The four limitations listed in Section 6 are real and appropriately scoped. The conformer quality gap between DFT pre-training structures and semi-empirically generated fine-tuning structures is a meaningful practical limitation that is not mentioned anywhere in the main text and should be.

**Strengths And Weaknesses:**

6-shot images achieve PCE MAE of 1.465 at 4.7 GiB GPU memory, while the best point cloud configuration reaches only 1.986 at 13.3 GiB. That is a meaningful gap, not a marginal one. The ablation in Table 8 is appropriately structured. Each of CAP, EGP, and QSP contributes independently, EGP contributes most, and the full combination is best. This is the kind of ablation the community can actually use.

The rotation-invariance proof in Appendix A is formally correct under the stated assumptions, but the implementation uses fixed canonical views while the theory assumes i.i.d. Haar sampling. Lemma A.4 patches this via Wasserstein discrepancy, but W_1(mu_K, mu) for K=6 is never computed, so the bound is numerically vacuous. Table 9 shows canonical and Fibonacci sampling differ by 0.001 MAE, well within reported variance, which does not validate the theory either way. Several gains in Tables 1 and 3 are real but weaker than the text suggests. In Table 1, VisionED improves VOC over VideoMol by 2.6% under the acceptor split (0.075 vs. 0.078, both std 0.002), which is within one standard deviation. No significance tests appear anywhere in the paper. The solvent scaffold gains in Table 3 are larger and more credible, particularly the 27.0% improvement on lambda-Abs, but the emitter split margins are modest throughout.

The ternary blend result in Table 4 is too thin to carry the weight placed on it. Five test systems, accuracy defined as 1 minus relative PCE deviation from a single experimental measurement, and no confidence intervals make this an anecdote. The 93.7% average accuracy claim in the abstract should not stand on five data points. The most consequential missing experiment is a standard ImageNet-pretrained ViT fine-tuned on the same downstream tasks. Table 8's w/o pre-training baseline is a randomly initialized ViT, which is not the right counterfactual. Without knowing how much lift comes specifically from ED-informed pre-training versus any pre-trained ViT backbone, the physics-informed framing cannot be fully evaluated. Section 2 cites VideoMol as an experimental baseline but does not engage with it as prior art for multi-view molecular image rendering, which it directly is. This gap in the related work misrepresents the originality of the rendering strategy.

---

> ### Author Rebuttal · Authors · 2026-03-30
>
> We appreciate your helpful feedback. We have responded to your concerns below and look forward to any additional comments.
> Due to character limitations, we have placed Tables R2-R6 in the link (https://anonymous.4open.science/r/Additional-Tables-3398/README.md).
>
> > Q1: Downstream performance of ImageNet-pretrained ViT
>
> A1: We would like to first clarify a misunderstanding regarding Table 8. The row with all pre-training tasks removed is not a randomly initialized ViT. Our encoder is built on ViT-base, and this ablation corresponds to the same ViT backbone initialized from standard ImageNet pre-training.  We have added Tables R2-R4, which compare VisionED against the ImageNet-pretrained ViT-base model across all three benchmarks. Specifically, compared with the ViT-base baseline, VisionED achieves average gains of 23.34% / 14.96% on OPEP2, 13.83% on HOPV15, and 19.51% / 31.86% on Deep4Chem.
>
> |$\Delta$|Voc|Jsc|FF|PCE|
> |-|-:|-:|-:|-:|
> |OPEP2 acceptor|16.67%|35.85%|12.79%|28.06%|
> |OPEP2 donor|12.05%|27.40%|7.08%|13.29%|
> |HOPV15|12.15%|18.10%|12.30%|12.76%|
> |$\Delta$|$\lambda_{Abs}$|$\lambda_{Emi}$|PLQY|FWHM|
> |Deep4Chem emitter|25.12%|17.35%|12.54%|23.01%|
> |Deep4Chem solvent|46.17%|41.43%|18.11%|21.74%
>
> > Q2: VisionED's gain over the best baseline
>
> A2: Across the 16 tasks in Tables 1 and 3, we find that VisionED exceeds one pooled standard deviation over the best baseline on 12/16 metrics. In addition, the pairwise t-tests show that 12/16 gains are statistically significant in Table R5.
>
> |p-value|Voc|Jsc|FF|PCE|
> |-|-:|-:|-:|-:|
> |OPEP2 acceptor|0.328|0.042|0.038|0.004|
> |OPEP2 donor|0.663|0.653|0.779|0.048|
> |p-value|$\lambda_{Abs}$|$\lambda_{Emi}$|PLQY|FWHM|
> |Deep4Chem emitter|9e-7|0.001|0.007|0.023|
> |Deep4Chem solvent|9e-15|2e-6|0.047|0.039|
>
> > Q3: Expanding to at least 15 systems and reporting variance
>
> A3: To address this concern, we have substantially expanded the ternary-blend experiment. Specifically, we increased the number of ternary systems from 5 to 15, and repeated the experiments with three random seeds in Table R6.  VisionED achieves an average Acc of 92.77%, which provides stronger evidence for the model’s capability in ternary-blend prediction.  Furthermore, we will revise the abstract and Section 5.2.4 to reflect the updated experiment and replace the previous 93.7% claim with the new result.
>
> |id|1|2|3|4|5|
> |-|-:|-:|-:|-:|-:|
> |Acc|95.92±3.27|97.32±3.17|88.22±4.78|88.43±3.55|86.48±6.19|
> ||6|7|8|9|10|
> ||98.08±5.19|95.44±5.44|96.77±4.81|95.44±3.84|90.89±3.16|
> ||11|12|13|14|15|
> ||93.41±6.43|95.54±3.26|80.13±4.69|94.00±1.53|95.47±3.28|
>
> > Q4: Distributional shift between pretraining and finetuning
>
> A4: To assess the potential distributional shift between the two protocols, we randomly sampled 1,000 molecules from the pre-training dataset and computed the ESSOR descriptors under both settings for comparison. Across the 69 ESSOR descriptors, the average MAE is 0.482±0.325, and the average Pearson correlation coefficient is 0.708±0.244. These results indicate that the two protocols retain a moderate level of ESSOR-level consistency.
>
> > Q5: Sampling strategy
>
> A5: In our practical pipeline, the relevant empirical measure is the distribution on SO(3) induced by the six orthogonal canonical views together with RandomRotation augmentation. The worst-case transport occurs at Voronoi-corner configurations, which yields the concrete bound $W_1({\mu}_6,\mu)\le \arccos(1/\sqrt3)$. We will include this explicit constant in the revision.
> Table 9 was not intended to validate the theorem; rather, it was included to study the effect of the sampling strategy under a fixed sampling budget. The result shows that, in a 6-shot setting, a more elaborate sampling pattern does not provide a clear practical advantage over our simple canonical design.
>
> > Q6: Rendering strategy & Conformer quality gap
>
> A6.1: We would like to clarify that our novelty is materially different from VideoMol. VideoMol uses multi-view rendering to capture 3D conformational details from molecular structures better. In contrast, our main motivation is to make ED practical and scalable with multi-shot images. As also noted by the other three reviewers, the idea is effective in reducing computational and storage costs.
>
> A6.2: We will therefore revise the manuscript to explicitly mention this issue in the main text.
>
> > Q7: The solvent scaffold gains credible, but emitter split are modest.
>
> A7: For $\lambda_{Abs}$, the mean within-group std is 16.2 when grouping by solvent, versus 2.9 when grouping by emitter. This suggests that $\lambda_{Abs}$ is substantially more sensitive to emitter variation, making emitter-scaffold split likely harder than solvent-scaffold split.
>
> We thank the reviewer for the constructive feedback, which has significantly improved our work. If our responses have addressed your concerns, we kindly ask you to consider raising your score. We remain open to further discussion if any points require additional clarification.

---

> > ### Author Rebuttal · Reviewer_N64c · 2026-04-04
> >
> > I have read the rebuttal. My original review posed questions to the authors, which the rebuttal has now addressed. I have posted my detailed post-rebuttal assessment as an Official Comment.

---

> > > ### Author Response · Authors · 2026-04-07
> > >
> > > Thank you very much for your thoughtful and detailed review. We sincerely appreciate the time and effort you devoted to carefully evaluating our paper. Your comments are highly valuable and have substantially strengthened our work. We will carefully incorporate all results and discussions into the revised version of the manuscript to ensure it meets the highest standards. Thank you again for your positive evaluation and follow-up feedback.

---

### Official Review · Reviewer_H9Uq · 2026-03-13

**Soundness:** 2
**Presentation:** 2
**Significance:** 3
**Originality:** 3
**Overall Recommendation:** 5
**Confidence:** 3

**Summary:**

The authors propose VisionED, which makes property predictions based on multi-view images of electron density. The model uses ViT backbone. The model is pre-trained on molecules from EDBench and finetuned on 3 downstream datasets, including HOPV15, OPEP2 and Deep4Chem.

**Compliance With Llm Reviewing Policy:**

Affirmed.

**Final Justification:**

My concerns are fully resolved during rebuttal.

**Key Questions For Authors:**

- Why not directly test on EDBench molecules but test on small downstream datasets instead?
- Did you pretrain the baseline models on the 2M EDBench data (e.g., the geometric graph models )?
- How are camera positions selected for the multi-shot images (what does orthographic mean)?

**Limitations:**

Yes.

**Strengths And Weaknesses:**

Strength

- While the idea of using multi-view image representation is not entirely new (e.g., in computer vision), extending it to electron density is effective in reducing computational and storage cost.
- The pertaining strategies are interesting and are helpful for performance.

Weakness:
- The description regarding data causes confusion. Did you run DFT calculations on the 2M pretraining set to obtain the electron density or did you directly use the electron density data from EDBench?
- The baseline models are not pertrained, which may lead to unfair comparison.
- Getting ED requires running DFT, which can be time-consuming.

---

> ### Author Rebuttal · Authors · 2026-03-30
>
> We appreciate your thorough review and helpful comments. We have carefully addressed your concerns below.
>
> > Q1: Did you run DFT calculations on the 2M pretraining set to obtain the electron density?
>
> A1: We apologize for the confusion. Specifically, EDBench provides the original ED in cube-file format, and we further convert these cube files into our multi-shot ED images using PyMOL. In addition, following the same basis-set setting as EDBench, we conduct large-scale DFT calculations to obtain the ESSOR descriptors. We will clarify this point in Sec. 4.2.
>
> > Q2: Why not directly test on EDBench molecules but test on small downstream datasets instead?
>
> A2: We focused our evaluation on HOPV15, OPEP2, and Deep4Chem for the following reasons.
>
> 1. Transferability to Real-world Properties: Our primary goal is to establish ED as a transferable representation for organic materials. Evaluating on experimentally grounded targets (photovoltaic and photophysical properties) better validates VisionED’s utility for material discovery.
> 2. Avoiding Data Leakage: Many EDBench quantum targets (e.g., energy) are already incorporated into our ESSOR descriptors and used as pre-training supervision. Using the pre-trained model for EDBench downstream tasks would overlap with our pre-training objectives, leading to an unfair evaluation.
> 3. While we prioritize downstream transfer, we already report validation and test performance on these quantum properties in Figure 9. VisionED achieves a high $R^2$ and low MAE, demonstrating strong performance on quantum tasks. For instance, on Total Energy, VisionED achieves an MAE of 9.82 a.u., an $R^2$ of 0.98, a HOMO-LUMO gap MAE of 0.04 a.u., and a Magnitude MAE of 0.55 a.u. on the test.
>
> > Q3: The baseline models are not pertrained, which may lead to unfair comparison.
>
> A3: In the current paper, we followed the official configurations of the baselines. For the geometric graph baselines, we used GeoFormer and EquiformerV2 as representative geometric graph models.  For GeoFormer, we adopt it as a competitive baseline because, in its original paper, it is already shown to outperform strong pretrained models.  For EquiformerV2, we used the official checkpoint trained on the S2EF-2M dataset for 30 epochs, where S2EF-2M is the 2M split of the OC20 S2EF dataset. We selected this checkpoint intentionally because its pre-training data scale matches our own 2M-molecule pretraining setting, and the task is to predict energy and forces from structure, making the comparison more directly comparable and fair.
>
> For the image-based baselines, ImageMol uses large-scale pre-training on 10M molecules, while VideoMol uses the same 2M pre-training data as ours. Across Tables 1-3, VisionED consistently outperforms VideoMol, which suggests that the gains of VisionED are not simply from using an image backbone or large-scale pre-training alone, but from the combination of ED representation and physics-informed supervision. To avoid ambiguity, we will revise the appendix to clearly state for each baseline, so that the comparison protocol is fully transparent.
>
> > Q4: Getting ED requires running DFT, which can be time-consuming.
>
> A4: Our work is not intended to reduce the cost of generating ED itself. Instead, the goal of VisionED is to provide a more efficient and transferable way to represent and learn from ED once it is available. By converting 3D ED into a compact multi-shot image, our method reduces storage cost and downstream modeling overhead while retaining useful electronic information. As discussed in the Appendix Sec E., we view the development of reliable and faster ED generation methods as an important future direction. To further investigate this issue, we added an exploratory experiment comparing ED generated by DeepDFT(DOI: 10.1038/s41524-022-00863-y) (an EGNN-based method for fast ED generation) and GFN2-xTB, as shown in Table R1. Although using DeepDFT-generated ED as input leads to slightly lower downstream performance than using higher-fidelity ED, it substantially reduces ED generation time. This preliminary result suggests a promising efficiency and accuracy trade-off.
>
> **Table R1. Comparison of DeepDFT and GFN2-xTB on the HOPV15 dataset. Total time refers to time of the ED generation for HOPV15.**
>
> | Model | DeepDFT | GFN2-xTB |
> | - | -: | -: |
> | PCE | 1.702±0.098 | 1.590±0.064 |
> | Total time/min | 5.3 | 20.5 |
>
> >  Q5: what does orthographic mean?
>
> A5: Our default multi-shot setting uses 6 canonical orthographic views: top, bottom, left, right, front, and back. The rendering process rotates the molecule along predefined axes and captures standardized views, as described in Section 4.2 and Table 7.
>
> We thank the reviewer for the constructive feedback, which has significantly improved our work. If our responses have addressed your concerns, we kindly ask you to consider raising your score. We remain open to further discussion if any points require additional clarification.

---

> > ### Author Rebuttal · Reviewer_H9Uq · 2026-04-05
> >
> > Thank you for the detailed rebuttal. My concerns are resolved. I would like to follow-up with some final confirmations:
> >
> > - To confirmation, some of the baseline models are pretrained but use pretraining data from different source or modality. Is this correct? How about other baselines in the paper?
> >
> > - Did you run fine-tuning for the baselines? If so, did you use similar procedures?

---

> > > ### Author Response · Authors · 2026-04-05
> > >
> > > Thank you for your follow-up questions. We are happy to clarify further the pretraining and fine-tuning settings of the baseline models.
> > > > Addition-Q1: To confirmation, some of the baseline models are pretrained but use pretraining data from different source or modality. Is this correct? How about other baselines in the paper?
> > >
> > > R1: Yes, this is correct: some baselines are pretrained, but their pretraining data and modalities differ. We summarize the detailed settings of all baselines in Additional Table R1. Specifically, EquiformerV2, ImageMol, and VideoMol are pretrained, using different pretraining sources or representations, while SVM, RF, PointVector, X-3D, and GeoFormer are trained from scratch without pretraining. Please note that VideoMol and VisionED use the same molecules for pre-training. The key difference lies in the input modality: VideoMol takes conformer videos as input, whereas VisionED uses multi-shot ED images.
> > >
> > > As discussed in Q3, GeoFormer is already a strong baseline and outperforms pretrained models in its original paper. Moreover, for 3D ED point cloud models, our additional results on HOPV15 further show that lack of pretraining is unlikely to be the main reason for their weaker performance. As shown in the Additional Table R2, VisionED-NonPretrain achieves a lower MAE than both X-3D and PointVector. Therefore, we did not introduce an additional pretraining stage for the 3D ED point cloud baselines and instead followed their original training protocols for fair comparison.
> > >
> > > > Addition-Q2: Did you run fine-tuning for the baselines? If so, did you use similar procedures?
> > >
> > > R2:  All methods underwent downstream train/validation/test on the same split. Pretrained baselines were initialized from pretrained weights, while the others were trained from scratch. To ensure fairness, we followed the original training or fine-tuning protocols for each method.
> > >
> > >
> > > **Additional Table R1:** The detailed summary of the baselines and our VisionED.
> > >
> > >
> > > | Category                | Method | Pretraining | Pretraining data              | Representation        | Downstream training                        | Hyperparameters | Training protocol         |
> > > |-------------------------|---|----------|----------------------------|-----------------------|-------------------------------------|---|--------------------------|
> > > | Traditional ML Models   | SVM | No       | -                          | ECFP descriptor       | Trained from scratch                | Hyperparameter search | Standard sklearn library |
> > > |                         | RF | No       | -                          | ECFP descriptor       | Trained from scratch                | Hyperparameter search | Standard sklearn library |
> > > | Point Cloud Models      | PointVector | No       | -                          | 3D ED point cloud     | Trained from scratch                | Grid search following the original paper | Following the original paper          |
> > > |                         | X-3D | No       | -                          | 3D ED point cloud     | Trained from scratch                | Grid search following the original paper | Following the original paper           |
> > > | Geometric Graph Models  | GeoFormer | No       | -                          | Geometric graph       | Trained from scratch                | Grid search following the original paper | Following the original paper           |
> > > |                         | EquiformerV2 | Yes      | 2M molecules from OC20          | Geometric graph       | Initialized from pretrained weights | Grid search following the original paper | Following the original paper           |
> > > | Visual Molecular Models | ImageMol | Yes      | 10M molecules from PubChem | Molecular topology image | Initialized from pretrained weights                                 | Grid search following the original paper | Following the original paper           |
> > > |                         | VideoMol | Yes      | 2M molecules from PCQM4Mv2 | Molecular conformer video | Initialized from pretrained weights                                 | Grid search following the original paper | Following the original paper           |
> > > |                         | VisionED | Yes      | 2M molecules from PCQM4Mv2 | Multi-shot ED images  | Initialized from pretrained weights                                 | Grid search |          Our implementation               |
> > >
> > >
> > > **Additional Table R2:** The MAE performance comparison on HOPV15 dataset.
> > >
> > > |Model|Voc|Jsc|FF|PCE|
> > > |---|---:|---:|---:|---:|
> > > |X-3D|0.138±0.019|3.868±0.298|10.087±0.927|1.800±0.136|
> > > |PointVector|0.121±0.013|4.095±0.419|10.599±0.678|1.986±0.093|
> > > |VisionED-NonPretrain|**0.107±0.008**|**2.856±0.195**|**8.442±0.510**|**1.465±0.081**|
> > >
> > > We thank the reviewer for the constructive feedback, which has significantly improved our work. If our responses have addressed your concerns, we kindly ask you to consider raising your score. We remain open to further discussion if any points require additional clarification.

---

### Decision · Program_Chairs · 2026-04-30

**Decision:**

Accept (regular)

**Comment:**

The submission proposes VisionED, an organic-material property prediction model with multi-view electron-density images. The key idea is to replace expensive electron-density point clouds with a more efficient image representation from multiple views, which could leverage prevalent and performant image processing architectures. The work also highlights a pre-training state on large-scale density-derived data. Reviewers agreed that the paper tackles an important problem and that the idea is practically relevant. The empirical study also received acknowledgement considering the multiple downstream datasets, low-data and OOD settings, ablations over pretraining objectives and view strategies, and efficiency comparisons against prior density point-cloud approaches.

The reviewers also raised several concerns, including not fully supported claims, novelty over previous density-aware methods, the cost of generating density inputs, and theoretical looseness in the rotation-invariance discussion. The authors provided substantial additional results that have addressed most concerns. While some still remain, the submission conveys noteworthy information and inspiration to the community overall and could inspire further works or findings.